# CALIBRATING EXPRESSIONS OF CERTAINTY

**Peiqi Wang** [*1]   **Barbara D. Lam** [2]   **Yingcheng Liu** [1]   **Ameneh Asgari-Targhi** [2]
**Rameswar Panda** [3]   **William M. Wells** [1,2]   **Tina Kapur** [2]   **Polina Golland** [1]
[1] CSAIL, MIT   [2] Harvard Medical School   [3] MIT-IBM Watson AI Lab

## ABSTRACT

We present a novel approach to calibrating linguistic expressions of certainty, e.g., "Maybe" and "Likely". Unlike prior work that assigns a single score to each certainty phrase, we model uncertainty as distributions over the simplex to capture their semantics more accurately. To accommodate this new representation of certainty, we generalize existing measures of miscalibration and introduce a novel post-hoc calibration method. Leveraging these tools, we analyze the calibration of both humans (e.g., radiologists) and computational models (e.g., language models) and provide interpretable suggestions to improve their calibration.

## 1 INTRODUCTION

Measuring the calibration of humans and computational models is crucial. For example, in healthcare, radiologists express uncertainty in natural language (e.g., "Likely pneumonia") due to the inherent ambiguity in the image they examine. These certainty phrases (e.g., "Maybe", "Likely") influence healthcare providers' decisions such as ordering additional diagnostic tests. Accurate perception of these certainty phrases directly impacts patient diagnosis and treatment. Additionally, it's more natural for large language models (LLMs) to express their confidence using certainty phrases since humans struggle with precise probability estimates (Zhang & Maloney, 2012). Our work enables measuring the calibration of both data annotators and LLMs, paving ways for future work to improve the reliability of LLMs.

Existing miscalibration measures focus on classifiers that provide a confidence score, e.g., posterior probability. These approaches cannot be applied directly to text written by humans or language models that communicate uncertainty using natural language. Prior work on "verbalized confidence" attempted to address this by mapping certainty phrases to fixed probabilities, e.g., "High Confidence" equals "90% confident", (Lin et al., 2022a). The oversimplification misses two key aspects: (1) individual semantics: people use phrases like "High Confidence" to indicate a range (e.g., 80-100%) rather than a single value; and (2) population-level variation: different individuals may interpret the same certainty phrase differently. Appendix D explains this gap in more detail.

Calibration in the space of certainty phrases presents unique challenges. Prior work such as histogram binning (Zadrozny & Elkan, 2001) and Platt scaling (Platt, 2000) fit low-dimensional functions (e.g., one-dimensional for binary classifiers) to map uncalibrated confidence scores to calibrated probabilities. However, when working with certainty phrases, direct manipulation of the underlying confidence scores is not feasible. Rather than mapping confidence scores directly, we instead calibrate a model by adjusting the use of different certainty phrases.

In this work, we measure and calibrate both humans and computational models that convey their confidence using natural language expressions of certainty. The key idea is to treat certainty phrases as distributions over the probability simplex. Using this construction, we generalize existing estimators for miscalibration metrics, such as the expected calibration error (ECE) (Pakdaman Naeini et al., 2015), and visualization tools, such as the reliability diagrams (Wilks, 2006). To calibrate over certainty phrases, we learn a discrete and possibly stochastic calibration map over a set of these certainty phrases to a potentially different set. This mapping is derived as the solution to an optimal transport problem that minimizes the net change in calibration error.

We demonstrate our approach by analyzing the calibration of radiologists writing clinical reports, accounting for variables such as the pathology and radiologist's identity. Moreover, we show how

---

[*]Correspondence to `wpq@mit.edu`; Code available on GitHub

we can guide radiologists to become better calibrated in their use of certainty phrases. In addition, we showcase the calibration of language models and demonstrate the effectiveness of our calibration method to post-hoc improve model calibration. Our research opens new avenues for assessing and improving the calibration of humans and computational models that communicate their confidence using natural language.

## 2 RELATED WORK

**Measuring Miscalibration**    We build our approach on the expected calibration error (ECE), a popular method for measuring miscalibration (Zadrozny & Elkan, 2001). Previous research has focused on adapting ECE to skewed confidence score distributions (Nguyen & O'Connor, 2015), improving its sample efficiency (Zhang et al., 2020), and debiasing (Roelofs et al., 2022). We extend the ECE estimator to accommodate the assumption that certainty phrases represent distributions rather than real-valued scores. This formulation is robust to binning strategies and alleviates estimation noise under small sample sizes (Kumar et al., 2019). Interestingly, our proposed ECE estimator can be viewed as a variant of kernel density estimator for ECE (Zhang et al., 2020; Popordanoska et al., 2022) that uses input-dependent, possibly asymmetric kernels.

**Calibrating Classifiers & Regressors**    We focus on post-hoc calibration, as opposed to training models to be calibrated ab initio. Post-hoc calibration reduces to estimating or fitting the canonical calibration function (Vaicenavicius et al., 2019). For example, histogram binning (Pakdaman Naeini et al., 2015) estimates the canonical calibration function with histogram regression. Platt scaling (Platt, 2000), isotonic regression (Zadrozny & Elkan, 2002), and Beta calibration (Kull et al., 2017) fit monotonic, potentially smooth functions from data. These methods assume models provide confidence scores, which makes them unsuitable when only certainty phrases are available.

Distribution calibration (Song et al., 2019) works on regressors that, akin to our setup, predict confidence distributions rather than scores. In contrast to distribution calibration, we focus on classification problems. Moreover, matching the predicted distribution to the true distribution of subsets of examples sharing the same prediction is not of interest in our application.

**Language Model Calibration**    Language model calibration typically involves defining real-valued confidence scores and using existing tools designed for calibrating classifiers. For instance, confidence scores can represent conditional probability of an answer given the context (Desai & Durrett, 2020) or the average probability across different paraphrases of answer tokens (Jiang et al., 2021). These methods require access to the model's internal state. Our work is closely related to "verbalized confidence" (Lin et al., 2022b; Tian et al., 2023), where the model articulates its confidence in token-space as certainty phrases. Rather than focusing solely on the mean, our method offers a more realistic quantification of calibration error by treating each certainty phrase as a distribution. Unlike previous studies that apply calibration techniques designed for classifiers (Desai & Durrett, 2020) or fine-tune the language model for controlled generation (Mielke et al., 2022), we propose a lightweight discrete policy that adjusts the use of certainty phrases to improve calibration.

## 3 METHOD

### 3.1 BACKGROUND

**Miscalibration Measures**    Let $\mathcal{X}$ be the input space and $\mathcal{Y}$ be a finite set of labels. For simplicity, we consider binary classification, i.e., $\mathcal{Y} = \{0, 1\}$. A probabilistic classifier $g : \mathcal{X} \to [0, 1]$ provides its confidence level for the positive class. Here, we use $g$ to refer to any type of classifier, whether it be a human or a computational model. Classifier $g$ is calibrated if the true probability of the positive class given model's prediction is exactly equal to that prediction:

$$\mathbb{E}\left[Y \mid S\right] = S, \tag{1}$$

where $S = g(X)$ represents the confidence score. For instance, a radiologist is perfectly calibrated if his predicted probabilities match real-world outcomes, e.g., if a radiologist predicts a 30% probability of pneumonia for a group of patients, then pneumonia actually occur in 30% of those patients.

Alternative definitions of calibration, e.g., confidence calibration (Guo et al., 2017) and class-wise calibration (Kull et al., 2019), are equivalent for binary classification (Vaicenavicius et al., 2019).

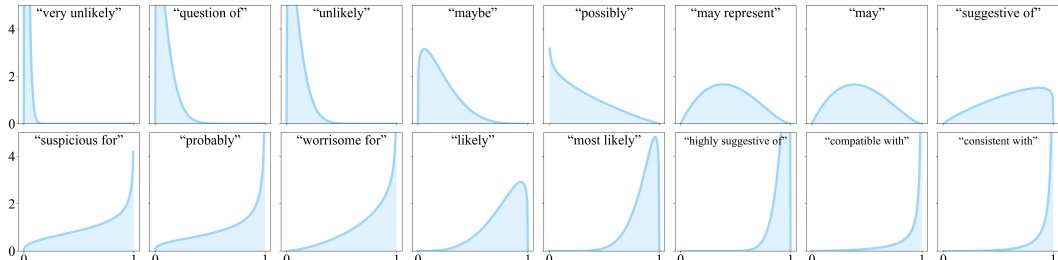

Figure 1: Probability density functions obtained by fitting beta distributions to results of a survey of radiologists' perception of different certainty phrases (Shinagare et al., 2023).

The expected calibration error (ECE) (Pakdaman Naeini et al., 2015) measures the degree to which the calibration definition is violated in expectation:

$$\text{ECE} = \mathbb{E}\left[\,\left|\mathbb{E}\left[Y \mid S\right] - S\right|\,\right], \tag{2}$$

where the outer expectation is computed with respect to the distribution $P_S$ that is the pushforward of $P_X$ by $g$. If $\text{ECE} = 0$, then $g$ is perfectly calibrated.

**Empirical Estimation**   Given a dataset $\mathcal{D} = \{(x_1, y_1), \cdots, (x_N, y_N)\}$, one can estimate ECE in Equation (2) or visualize the reliability diagram (DeGroot & Fienberg, 1983) with binning (Pakdaman Naeini et al., 2015). Specifically, we use $(I_1, \cdots, I_M)$ to denote the set of bins that partition classifier $g$'s range and $B_m$ to denote the set of indices of samples whose predictions fall into bin $I_m$. Empirical averages

$$\hat{r}_m = \frac{1}{|B_m|} \sum_{n \in B_m} y_n \qquad \hat{p}_m = \frac{|B_m|}{N} \qquad \hat{g}_m = \frac{1}{|B_m|} \sum_{n \in B_m} g(x_n) \tag{3}$$

serve as estimates for per-bin calibration function $r_m \triangleq \mathbb{E}\left[Y \mid S \in I_m\right]$, score probability $p_m \triangleq P(S \in I_m)$, and model prediction $g_m \triangleq \mathbb{E}\left[S \mid S \in I_m\right]$ respectively.

The binning estimator for ECE is a weighted average of per-bin calibration errors

$$\widehat{\text{ECE}} = \sum_{m=1}^{M} \hat{p}_m \left|\hat{r}_m - \hat{g}_m\right|. \tag{4}$$

Statistical properties of the estimator above have been studied in (Vaicenavicius et al., 2019).

If we define $\hat{r}(s) = \hat{r}_m$ for $s \in I_m$, function $\hat{r}$ is the histogram regression estimator for the canonical calibration function $r(s) \triangleq \mathbb{E}\left[Y \mid S = s\right]$. The reliability diagram displays the curve $(s, \hat{r}(s))$ for $s \in [0, 1]$. If $g$ is perfectly calibrated, then $\hat{r}(s) = s$ for $s \in [0, 1]$.

**Post-hoc Calibration**   Given a model $g$ and a calibration dataset $\mathcal{D}$, post-hoc calibration seeks a calibration map $t : [0, 1] \to [0, 1]$ such that $t \circ g$ is well-calibrated. This can be achieved by minimizing $\mathbb{E}_{(X,Y) \sim \mathcal{D}} \ell(t(g(X)), Y)$ over a family $\mathcal{T}$ of possible mappings, where $\ell$ is a loss function. Platt scaling (Platt, 2000) is an example of this approach, where $\mathcal{T}$ is the set of logistic functions and $\ell$ is the log-loss.

## 3.2   Confidence as a Distribution

Let $\mathcal{U} \triangleq \{u_1, \cdots, u_K\}$ be a set of $K$ distributions corresponding to different certainty phrases. For instance, the phrase "Maybe" could be modeled as $\text{Beta}(2, 2)$, which is centered at $0.5$ with decaying density away from $0.5$. Figure 1 illustrates what these distributions might look like. We can derive confidence distributions from surveys on human perception of certainty phrases, e.g., in radiology (Shinagare et al., 2023) or from social media polls (Fagen-Ulmschneider, 2023). In this work, we employ beta distributions to represent the confidence distributions $u_1, \cdots, u_K$.

We assume that instead of outputting a scalar-valued score, the classifier $g : \mathcal{X} \to \mathcal{P}([0, 1])$ provides its confidence for the positive class as a distribution over $[0, 1]$. Here, we use $g$ to abstract a human

or a language model that generates certainty phrases about its internal belief of an event, e.g., a radiologist might dictate "Likely pneumonia." To restrict our attention to a finite set of certainty phrases, we assume $g = u \circ \gamma$ where $\gamma : \mathcal{X} \to [K]$ outputs an index and $u : [K] \to \mathcal{P}([0, 1])$ maps that index to the corresponding confidence distribution, i.e., $u(k) = u_k$. Given any sample $x \in \mathcal{X}$, the classifier's output $g(x) = u_{\gamma(x)}$ is one of the $K$ confidence distributions. Alternatively, $g(x)$ can be interpreted as defining a conditional density for the score $S$ that is a finite mixture over $\{u_k\}$, i.e.,

$$f_{S|X}(s \mid x) = \sum_{k=1}^{K} \mathbb{1}\left[\gamma(x) = k\right] u_k(s). \tag{5}$$

The definitions of calibration and ECE in Equations (1) and (2) remain valid even as we move to treat confidence as a distribution. Instead of being the pushforward of $P_X$ by $g$, $P_S$ is characterized by the density $f_S(s) = \mathbb{E}\left[f_{S|X}(s \mid X)\right]$. We generalize estimators in Equation (3) as follows:

$$\hat{r}_m = \frac{\sum_{n=1}^{N} P(S \in I_m \mid X = x_n) y_n}{\sum_{n=1}^{N} P(S \in I_m \mid X = x_n)} \qquad \hat{p}_m = \frac{1}{N} \sum_{n=1}^{N} P(S \in I_m \mid X = x_n) \tag{6}$$

$$\hat{g}_m = \frac{\sum_{n=1}^{N} P(S \in I_m \mid X = x_n) \mathbb{E}\left[S \mid S \in I_m, X = x_n\right]}{\sum_{n=1}^{N} P(S \in I_m \mid X = x_n)}. \tag{7}$$

**Proposition 1.** *The estimators $\hat{r}_m$, $\hat{p}_m$, and $\hat{g}_m$ defined in Equation (6) are consistent estimators for $\mathbb{E}\left[Y \mid S \in I_m\right]$, $P(S \in I_m)$, and $\mathbb{E}\left[S \mid S \in I_m\right]$ respectively.*

We provide the proof in Appendix A.1.

**Interpretation** The estimators in Equation (6) reduce to that in Equation (3) if the confidence distributions are delta distributions, i.e., $\mathcal{U} = \{\delta_s \mid s \in [0, 1]\}$. Provided that the confidence distributions are supported on $[0, 1]$, we can interpret the estimators in Equation (6) as allowing each sample to contribute to the estimates $\hat{r}_m, \hat{g}_m, \hat{p}_m$ of every bin $I_m$ with a smaller weight determined by $P(S \in I_m \mid X = x_n)$. The resulting estimates are more robust to increasing the number of bins and will not result more biased calibration errors as we explain in Section 3.3.

**Implementation** Since the confidence distributions are represented via their parametric form, computing $P(S \in I_m \mid X = x_n)$ is straightforward and requires two evaluations of the CDF of $u_{\gamma(x_n)}$. In contrast, computing $\mathbb{E}\left[S \mid S \in I_m, X = x_n\right]$ involves integrating w.r.t. a density. If the number of bins $M$ is small, we can use adaptive quadrature algorithms through `scipy.integrate` to trade time for better approximation; If $M$ is large, midpoint rule provides sufficiently good approximation of the integral, requiring a single evaluation of the density $u_{\gamma(x_n)}$.

**Accounting for Uncertain Labels** In practice, ground truth labels $y_1, \cdots, y_n$ may not always be available. Instead, we might only have access to certainty phrases that provide information about the latent binary labels. We can incorporate such uncertain labels into calibration error estimation, which is especially useful when the sample size is small and many labels are uncertain. Let $c_n \in \mathcal{P}([0, 1])$ be the confidence distribution associated with the certainty phrases for label $y_n$. To account for this uncertainty, we can replace $y_n$ with $P(c_n \geq \frac{1}{2})$ in the estimator $\hat{r}_m$ defined in Equation (6). The modified estimator remains consistent that we prove in Appendix A.2.

## 3.3 Connection to Kernel Density Estimation of ECE

Suppose $I_1, \cdots, I_m$ are equal width intervals, i.e., $I_m = (s_m, s_m + \delta)$ for some $\delta > 0$. We define continuous versions of the estimator $\hat{r}_m$ and $\hat{p}_m$ in Equation (6)

$$\hat{r}(s) = \frac{\sum_{n=1}^{N} f_{S|X}(s \mid x_n) y_n}{\sum_{n=1}^{N} f_{S|X}(s \mid x_n)} \qquad \hat{f}_S(s) = \frac{1}{N} \sum_{n=1}^{N} f_{S|X}(s \mid x_n) \tag{8}$$

by considering the limit when the number of bins $M$ goes to infinity. Specifically,

$$\hat{r}_m = \frac{\sum_{n=1}^{N} P(S \in I_m \mid X = x_n) y_n}{\sum_{n=1}^{N} P(S \in I_m \mid X = x_n)} \approx \frac{\sum_{n=1}^{N} \delta f_{S|X}(s_m \mid x_n) y_n}{\sum_{n=1}^{N} \delta f_{S|X}(s_m \mid x_n)} = \hat{r}(s_m), \tag{9}$$

as $P(S \in I_m \mid X = x_n) \approx \delta f_{S|X}(s_m \mid x_n)$ when $M \to \infty$ or $\delta \to 0$.

For each sample $x_n \in \mathcal{X}$, let $\alpha_n, \beta_n$ be parameters of the predicted confidence distribution $g(x_n) \sim$ Beta$(\alpha_n, \beta_n)$. We can define an input-dependent kernel $K_{h_n}(s - s_n) = f_{g(x_n)}(s)$ where $s_n \triangleq \frac{\alpha_n - 1}{\alpha_n + \beta_n - 2}$ is the "mode" and $h_n \triangleq \frac{1}{\alpha_n + \beta_n - 2}$ is the "spread" of the kernel. Note

$$\hat{r}(s) \equiv \frac{\sum_{n=1}^N K_{h_n}(s - s_n) y_n}{\sum_{n=1}^N K_{h_n}(s - s_n)} \tag{10}$$

is the Nadaraya–Watson estimator (Nadaraya, 1964; Watson, 1964) for the canonical calibration function $r(s) = \mathbb{E}[Y \mid S = s]$. This estimator is akin to the KDE-based estimator proposed in (Zhang et al., 2020; Popordanoska et al., 2022). However, unlike the KDE-based estimator where $s_n$ is the predicted confidence score and $h_n$ is a fixed hyperparameter common to all $n$, in the estimator in Equation (10), both $s_n$ and $h_n$ are parameters induced by the confidence distribution $g(x_n)$.

We define a continuous version of the estimator for ECE in Equation (2) as

$$\widetilde{\text{ECE}} = \int |\hat{r}(s) - s| \, \hat{f}_S(s) \, ds. \tag{11}$$

The estimate $\widehat{\text{ECE}}$ in Equation (4) is a finite-sample approximation of $\widetilde{\text{ECE}}$. $\widetilde{\text{ECE}}$ can be shown to be unbiased and consistent (Zhang et al., 2020; Popordanoska et al., 2022).

**Caveats** When the model predicts delta distributions as confidence distributions, e.g., $g(x_n) \sim \delta_s$ for some $s \in [0, 1]$ indicating "100% Confident," special attention is required. These samples can be easily missed when constructing estimates of $\hat{r}(\cdot)$ and $\hat{f}_S(\cdot)$ if the sequence $s_1, \cdots, s_M$ is not properly chosen. For continuous confidence distributions $\mathcal{U}$, we approximate $P(S \in I_m \mid X = x_n)$ using the midpoint rule when computing the estimators in Equation (6). This numerical approximation of $\hat{r}_m$ is exactly $\hat{r}(\cdot)$ defined in Equations (8). If some predicted confidence distributions $g(x_n)$ are delta distributions, we compute the binning estimators in Equation (6) for the bin $I_m$ where $s \in I_m$ without any approximation.

## 3.4 Obtaining Calibration Maps Using Optimal Transport

**Calibration Map** If a classifier $g$ outputs confidence distributions, we do not have direct access to the confidence scores, rendering the classical definition of a calibration map $t : [0, 1] \to [0, 1]$ inapplicable. A natural extension is to consider map $t : \mathcal{P}([0, 1]) \to \mathcal{P}([0, 1])$ that adjusts confidence distributions in arbitrary ways. However, this definition complicates the interpretation of results. For instance, if Beta$(2, 2)$ represents "Maybe", it is unclear what a slightly modified distribution, such as Beta$(2, 2.2)$, represents in natural language.

Instead, we define a calibration map $t : [K] \to [L]$ that maps elements of the source set of $K$ confidence distributions $\{u_1, \cdots, u_K\}$ to elements of a target set of $L$ confidence distributions $\{v_1, \cdots, v_L\}$. More concretely, let $T \in \mathbb{R}_+^{K \times L}$ denote the transport matrix, where $T_{kl}$ represents the proportion of samples for which the confidence is better described by $v_l$ rather than $u_k$. We then define $t(k)$ as a draw from the categorical distribution with unnormalized probabilities $(T_{k1}, \cdots, T_{kL})$. For example, if $u_k$ and $v_l$ are confidence distributions that correspond to "Likely" and "Maybe" respectively, then $T_{kl} / \sum_l T_{kl} = 0.3$ indicates that the model should change 30% of its use of "Likely" to "Maybe" to reduce overconfidence.

**Calibration as Discrete Optimal Transport** Given that model $g$ is defined as $u \circ \gamma$, post-hoc calibration aims to find $t$ such that the composition $u \circ t \circ \gamma$ is well-calibrated. To achieve this, we formulate a discrete optimal transport problem. Let $a_k = \frac{1}{N} \sum_{n=1}^N \mathbb{1}[\gamma(x_n) = k]$ be the proportion of times $k$-th certainty phrase is mentioned. Therefore, the source weight $a = (a_1, \cdots, a_K) \in \triangle^{K-1}$ represents occurrences of certainty phrases in the calibration dataset. In some application, we are given target weights $b = (b_1, \cdots, b_L) \in \triangle^{L-1}$ that represent the ideal use of the target confidence distributions. We define a cost matrix $C \in \mathbb{R}^{K \times L}$ where

$$C_{kl} = \frac{1}{a_k} \left( \widehat{\text{ECE}}(u_k \to v_l) - \widehat{\text{ECE}} \right) \tag{12}$$

is the per-unit net change in the calibration error when we replace $u_k$ with $v_l$ as the confidence distribution. If target weight $b$ is given, we can obtain the calibration map $t$ by solving the classic Kantorovich formulation of discrete optimal transport (Peyré & Cuturi, 2019) that minimizes $\langle C, T \rangle$ over $T \in \mathbb{R}^{K \times L}$ subject to the constraints $T\mathbf{1} = a$ and $T^T\mathbf{1} = b$. The optimal transport plan $T$ minimizes the amount of work, e.g., change in the expected calibration error, required to transform $\sum_k a_k u_k$ to $\sum_l b_l v_l$. In practice, the target weight $b$ is often unknown and that we use the same source and target confidence distributions. In this case, it is generally desirable to ensure that the transport plan $T$ does not significantly change the relative use of source certainty phrases. To achieve this, we use unbalanced optimal transport (Frogner et al., 2015; Peyré & Cuturi, 2019):

$$\min_{T \in \mathbb{R}_+^{K \times L}} \langle C, T \rangle + \epsilon H(T) + \tau_1 \mathrm{KL} \left( T\mathbf{1} \| a \right) + \tau_2 \mathrm{KL} \left( T^T\mathbf{1} \| a \right) \tag{13}$$

where $H(T) = -\langle T, \log(T) \rangle$ is the discrete entropy of $T$ and $\tau_1, \tau_2$ controls the degree of marginal deviations. In practice, we set a large $\tau_1$ to strictly enforce the constraint $T\mathbf{1} = a$ while a smaller $\tau_2$ to permit some deviation of transported mass $T^T\mathbf{1}$ from the initial proportions $a$.

## 4    EXPERIMENTS

### 4.1    IMPLEMENTATION DETAILS

**Evaluation Metrics**    We estimate ECE as described in Section 3.2. Specifically, we partition $[0, 1]$ into 100 equal-width bins and compute the estimators in Equation (6) for each bin. To improve computational efficiency, we apply the midpoint rule to approximate the integrals, except for the first and last bins to account for $g(x_n) \in \{\delta_0, \delta_1\}$ for some $x_n$. We use bootstrap resampling with 100 samples to calculate the mean and 95% confidence interval for these estimators. In addition to ECE, we report a modified calibration error, ECE*, which excludes 100% confident predictions (e.g., $\delta_0, \delta_1$) in the first and last bins, and therefore better characterizes the calibration curves that are unaffected by extremely confident predictions. To compute classification statistics, such as the Brier Score (BS) and accuracy (Acc). We also employ bootstrap resampling, with further sampling from the predicted confidence distributions $g(x_n)$ and the ground truth confidence distributions $c_n$, to obtain the scalar-valued scores and binary labels required to compute these metrics.

**Optimal Transport Calibration**    We solve the entropy-regularized unbalanced transport problem in Equation (13) using the log-stablization variant of the Sinkhorn algorithm (Cuturi, 2013; Chizat et al., 2018; Schmitzer, 2019) to mitigate numerical instabilities. We use POT's solver implementation (Flamary et al., 2021). Based on ablation studies (Appendix B.5), we set $\epsilon = $ 1e-3 to minimize mass splitting and simplify interpretation of the calibration map. By default, we set $\tau_2 = $ 1e-3 arbitrarily due to its minimal impact on performance given such a small $\epsilon$.

### 4.2    CALIBRATING RADIOLOGISTS

To demonstrate the utility of our approach, we analyze the calibration of radiologists on a dataset of reports dictated for X-ray images and matching CT scans.

**Dataset**    We curated a dataset of 2,662 paired chest X-ray and CT radiology reports recorded from January to September 2023 at Brigham and Women's Hospital. We treat X-ray reports as human predictions $\{g(x_1), \cdots, g(x_N)\}$ and CT reports as ground truth labels $\{y_1, \cdots, y_N\}$ because the 3D CT scans provide greater details. Both types of reports contain certainty phrases about various pathologies. Appendix B.2 provides an example X-ray report. To mitigate distribution shifts in the use of certainty phrases, we use stratified sampling and split the dataset equally into calibration and test sets. We anonymize the names of radiologists with commonly used names in the US (Remy, 2021). To extract (pathology, confidence) labels from clinical reports, we prompt Llama 3 8B (Dubey & et al., 2024) with in-context learning. This approach demonstrates good accuracy on a hand-annotated test set of about 150 X-ray reports, but performance is slightly lower on the CT reports, likely due to their more detailed descriptions. Appendix B.4 provides further details.

**Confidence Distributions**    We derive the confidence distributions $\{u_1, \cdots, u_K\}$ from survey data on radiologists' interpretation of "diagnostic certainty phrases" commonly used in dictating radiol-

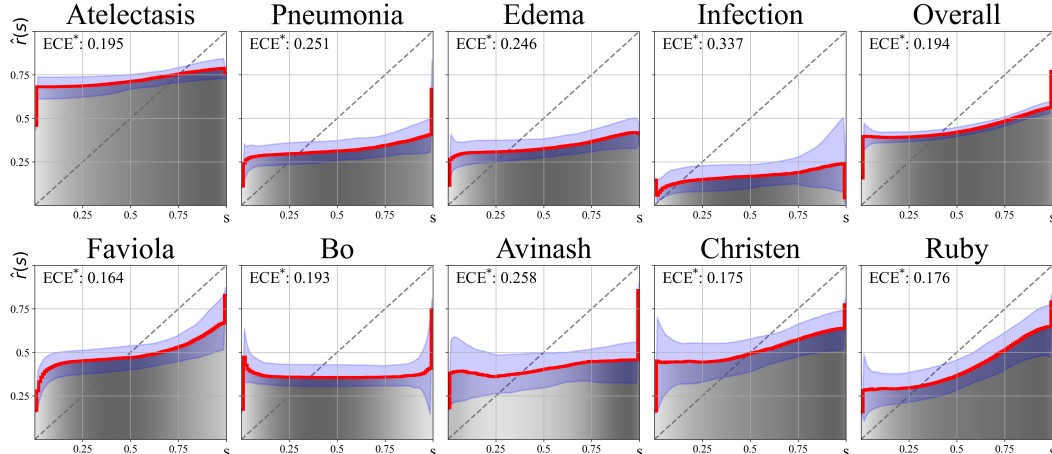

Figure 2: Reliability diagrams of radiologists' certainty phrase use in clinical reports, stratified by pathology (top) and radiologist identity (bottom). The calibration curve (red), with its 95% confidence interval (blue) and score density (gray) are shown. There is significant variation in calibration across different pathologies and radiologists. Areas where the calibration curve is above the identity line correspond to radiologists underestimating their confidence. Interestingly, this correlates with regions of low confidence. Same is true about overestimation in regions of high confidence.

ogy reports (Shinagare et al., 2023). Specifically, we fit beta distributions to the empirical data using the method of moments. Appendix B.3 provides further details.

**Measuring Radiologists' Calibration**    Figure 2 illustrates the variation in radiologists' calibration across different pathologies and between individual providers. Radiologists are generally underconfident in diagnosing common pathologies like atelectasis but tend to be overconfident with more ambiguous conditions like infection. In general, they exhibit underconfidence when expressing low certainty (e.g., using terms like "Possibly") and overconfidence with high certainty (e.g., "Likely"). This is consistent with prior studies on human perception of probabilities (Tversky & Kahneman, 1992). Moreover, individual radiologists display unique calibration patterns and distinct behaviors in their use of certainty phrases. Individual preferences for certain terms can contribute to differences in calibration. For instance, Bo frequently uses the term "May" while Christen often uses "Likely." Avinash is less calibrated than Bo, despite sharing similar calibration curves.

**Improving Human Calibration**    Figure 3 demonstrates the effectiveness of our calibration method in providing interpretable recommendations to improve radiologists' calibration. For instance, the calibration map suggests that radiologists use "May" instead of "Absent" to address underconfidence when diagnosing atelectasis. Similarly for edema diagnosis, the calibration map recommends lowering the confidence by replacing "Present" and "Likely" with "May". By following these straightforward adjustments, our method reduces the $ECE^*$ and Brier Score (BS) on the test set. In contrast, classical methods like histogram binning or Platt scaling does not provide clear guidance on how radiologists can adjust their language in reporting to improve calibration.

After calibration, the calibration curve is not perfectly aligned with the identity function for two reasons. First, adjustments to a small and discrete set of certainty phrases inherently limit precision in fine-tuning the curve. Second, the improvement in calibration depends on model's discriminative performance. For instance, the left side of the calibration curve (top-right subfigure in Figure 3) cannot be lowered further without a model that more accurately detects the absence of atelectasis.

In Appendix C.1, we examine the effect of $\tau_2$ on the resulting calibration map, illustrating the tradeoff between preserving assessment informativeness and improving calibration.

## 4.3    CALIBRATING LANGUAGE MODELS

To demonstrate the utility of our approach, we analyze the calibration of language models (LMs) on question-answering datasets.

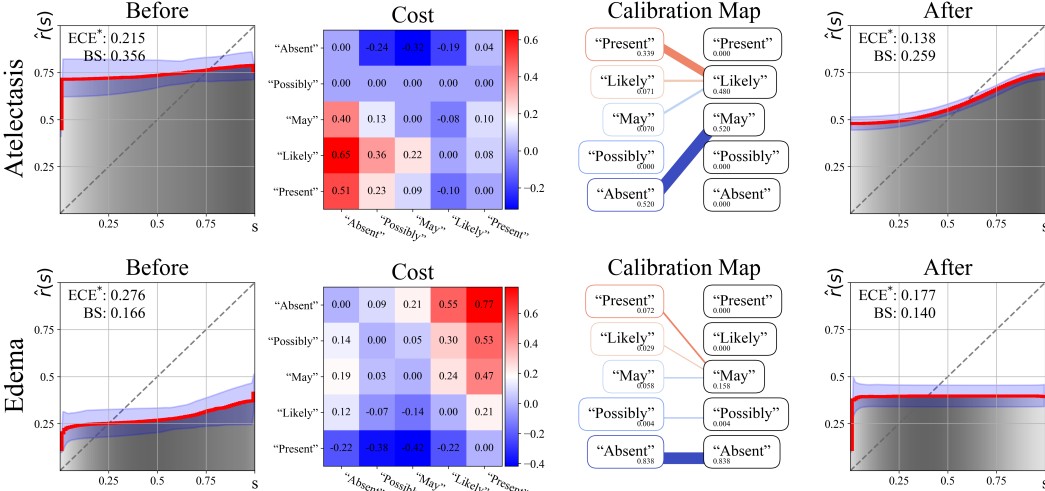

Figure 3: Examples of calibrating radiologists on two representative pathologies: atelectasis and edema. The 1st and 4th columns show the reliability diagrams before and after the post-hoc calibration, respectively. The 2nd column displays the cost matrix $C$ of the optimal transport problem, while the 3rd column illustrates the probabilistic calibration map $T$. For atelectasis, underconfidence can be addressed by suggesting the use of "May" instead of "Present"; For edema, overconfidence can be mitigated by recommending that radiologists replace "Present" and "Likely" with "May". Quantitatively, our calibration approach improves ECE and Brier Score (BS) metrics.

**Models**  We evaluate state-of-the-art LMs, such as gpt-4o (gpt-4o-2024-08-06), claude-3.5-sonnet (claude-3-5-sonnet-20240620), and gemini-1.5-pro (gemini-1.5-prob-002), as well as their smaller and faster variants, such as gpt-4o-mini (gpt-4o-2024-07-18), claude-3-haiku (claude-3-haiku-20240307), and gemini-1.5-flash (gemini-1.5-flash-002).

**Evaluation Setup**  Following Tian et al. (2023), we use two question-answering datasets: (1) SciQ (Welbl et al., 2017) contains crowd-sourced science exam questions, and (2) TruthfulQA (Lin et al., 2022c) contains questions designed to test language models' tendency to mimic human misconceptions. We use all 1,000 questions for SciQ and 817 questions for TruthfulQA from their respective validation set. Each dataset is evenly split into calibration and test sets using stratified sampling.

We focus on evaluating language models' ability to verbalize their confidence in natural language (Lin et al., 2022a). We prompt the models to provide either a probability (e.g., "0.3" or "0.7"), a certainty phrase (e.g., "Maybe", "Likely"), or a distribution directly (e.g., "Beta(2, 3)"). Appendix B.2 includes an example of how LMs express confidence using certainty phrases.

Similar to Tian et al. (2023), we derive ground truth labels by using gpt-4o-mini to evaluate whether a model's answer is semantically equivalent to the correct answer, thereby avoiding false negatives that arise from exact matching. For the TruthfulQA dataset, we verify whether the predicted answer matches any of the correct answers to further reduce false negative rates.

Appendix B.6 provides more details on prompts used. Table 5 lists all prompt templates.

**Confidence Distributions**  We investigate how the choice of certainty phrases and their corresponding distributions $u_1, \cdots, u_K$ impacts calibration performance. We prompt gpt-4o to generate $K$ pairs of (certainty phrase, distribution) using the prompt template in last row of Table 5. Figure 7 shows that calibration performance varies with $K$, but remains consistent across models and datasets. Based on this analysis, we choose $K = 12$ for subsequent experiments.

Table 3 compares calibration performance when confidence distributions are derived by: (1) *survey*: fitting beta distributions to survey data on human perceptions of probability-related terms (Fagen-Ulmschneider, 2023), (2) *on-the-fly*: generating beta distribution parameters for each sample, and (3) *fixed*: selecting from a predefined set of certainty phrases generated by gpt-4o. We find that pre-

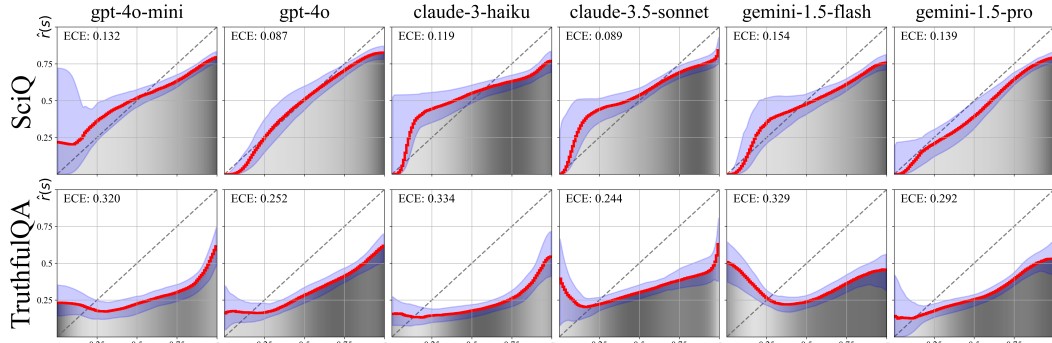

Figure 4: Reliability diagrams of LMs verbalizing confidence from a fixed set of certainty phrases generated by prompting gpt-4o, evaluated on SciQ (top) and TruthfulQA (bottom). The calibration curve (red), with its 95% confidence interval (blue), and score density (gray) are shown. Models are better calibrated on SciQ than TruthfulQA, with larger models (e.g., gpt-4o) outperforming their smaller variants (e.g., gpt-4o-mini). The smooth calibration curve improves the ability of human viewers to compare the calibration performance of different models.

defining a fixed set of phrases for the model to choose from yields better calibration than generating confidence distributions on-the-fly. Allowing the model to propose its own certainty phrases and rely on its perception of these phrases proves more effective than using those provided by humans.

Appendix B.3 provides additional details. Table 2 provides visualization of the probability density functions of the confidence distributions studied.

**Measuring Language Models' Calibration** Figure 4 illustrates the calibration performance of various LMs verbalizing confidence by selecting from a set of certainty phrases generated by prompting gpt-4o. Table 5 (row 5) provides a detailed view on the certainty phrases used. Current state-of-the-art language models exhibit good calibration on the SciQ dataset but perform less well on the TruthfulQA dataset. Larger models, such as gpt-4o, consistently outperform their smaller counterparts, like gpt-4o-mini, in terms of calibration. Models from the same family (e.g., gpt-4o and gpt-4o-mini) exhibit more similar characteristics compared to models from different vendors. For instance, gpt-4o tends to be more confident than claude-3.5-sonnet, even though both models achieve similar calibration errors.

Appendix C.3 demonstrates that our method yields a smooth calibration curves that enables clearer differentiation between model calibration profiles and better correspondence to values of ECE. Unlike binned calibration curves, which are sensitive to bin size and harder to interpret, our approach produces stable, consistent curves robust to binning variations.

**Post-hoc Calibration of Language Models** We compare our optimal transport calibration method with two classic calibration methods: Platt scaling (Platt, 2000) and histogram binning (Zadrozny & Elkan, 2001). These baseline methods are evaluated in two scenarios: (1) verbalized probability and (2) verbalized certainty phrases. For the second scenario, since the baseline methods only work with scalar confidence scores, we mapped the verbalized phrases to the mean of their corresponding confidence distributions. (Tian et al., 2023) also evaluates histogram binning under the verbalized phrase setup. To ensure a fair comparison, we calculate ECE and Brier Score (BS) for our method the same way as for the baselines, as opposed to following our formulation. Our method is unique in producing calibrated certainty phrases directly, while baselines require conversion to scalar confidence scores during calibration.

Table 1 demonstrates the effectiveness of our optimal transport calibration method in improving the calibration of language models. We show that even after reducing the output natural language expressions of certainty to scalar values (by simply taking the mean of the corresponding confidence distribution), our method remains competitive and does not compromise performance in terms of accuracy. We emphasize that our method's key advantage isn't in outperforming baseline calibration methods, but in directly operating on and producing natural language certainty phrases - a prop-

Table 1: Comparison of post-hoc calibration methods for language models expressing confidence as probabilities or certainty phrases. Our optimal transport calibration method, even after reducing output certainty phrases to scalar values (e.g., by taking the mean of the confidence distribution), remains competitive with calibration baselines without compromising accuracy. Unlike traditional methods limited to confidence scores, our approach directly operates on and produces certainty phrases, offering actionable suggestions to improve human calibration.

| Model | Verbalize | Calibrated? | SciQ | | | TruthfulQA | | |
|---|---|---|---|---|---|---|---|---|
| | | | Acc ↑ | ECE ↓ | BS ↓ | Acc ↑ | ECE ↓ | BS ↓ |
| gpt-4o | probability | ✗ | 0.72 | 0.18 | 0.22 | 0.3 | 0.52 | 0.43 |
| | | ✓ (scaling) | 0.72 | 0.11 | 0.19 | 0.3 | 0.11 | 0.2 |
| | phrase | ✗ | 0.73 | 0.07 | 0.19 | 0.35 | 0.22 | 0.28 |
| | | ✓ (scaling) | 0.73 | 0.09 | 0.17 | 0.35 | 0.11 | 0.2 |
| | | ✓ (binning) | 0.73 | 0.07 | 0.17 | 0.35 | 0.05 | 0.2 |
| | | ✓ (ours) | 0.73 | 0.08 | 0.18 | 0.35 | 0.1 | 0.21 |
| claude-3.5-sonnet | probability | ✗ | 0.69 | 0.24 | 0.24 | 0.34 | 0.42 | 0.37 |
| | | ✓ (scaling) | 0.69 | 0.1 | 0.19 | 0.34 | 0.03 | 0.2 |
| | phrase | ✗ | 0.66 | 0.06 | 0.22 | 0.33 | 0.25 | 0.3 |
| | | ✓ (scaling) | 0.66 | 0.06 | 0.21 | 0.33 | 0.06 | 0.21 |
| | | ✓ (binning) | 0.66 | 0.06 | 0.21 | 0.33 | 0.04 | 0.21 |
| | | ✓ (ours) | 0.66 | 0.07 | 0.22 | 0.33 | 0.04 | 0.21 |
| gemini-1.5-pro | probability | ✗ | 0.72 | 0.22 | 0.24 | 0.27 | 0.5 | 0.46 |
| | | ✓ (scaling) | 0.72 | 0.03 | 0.19 | 0.27 | 0.12 | 0.19 |
| | phrase | ✗ | 0.68 | 0.17 | 0.22 | 0.34 | 0.28 | 0.31 |
| | | ✓ (scaling) | 0.68 | 0.09 | 0.19 | 0.34 | 0.04 | 0.2 |
| | | ✓ (binning) | 0.68 | 0.07 | 0.19 | 0.34 | 0.07 | 0.2 |
| | | ✓ (ours) | 0.68 | 0.06 | 0.19 | 0.34 | 0.07 | 0.21 |

erty that existing methods lack. This makes our approach uniquely suited for improving human calibration in real-world settings, as it provides actionable guidance (e.g., suggesting radiologists use "May" instead of "Present" in their reporting to mitigate overconfidence) rather than abstract probability adjustments that can't be easily understood by humans.

Figure 13 provides visualizations of the calibration process for different language models. While the calibration maps follow a similar overall trend, they reveal distinct differences based on each model's distinct preference for certainty phrases.

## 5 DISCUSSIONS

**Limitations and Future Work**  Our study demonstrates that radiologists can improve their calibration by strictly following our proposed calibration method. However, it remains to be investigated how receptive radiologists are to calibration-improving suggestions and whether they can mentally adjust their use of certainty phrases effectively. Conducting a clinical study to assess these behavioral aspects would provide valuable insights into the practical benefits of our work.

**Conclusions**  This work presents a novel approach to modeling certainty phrases as probability distributions instead of fixed confidence scores. By adopting this perspective, we generalize existing estimators for ECE and introduce a smooth calibration curve for reliability diagrams. Additionally, we propose an interpretable post-hoc calibration method based on optimal transport that provides actionable calibration maps. Our method effectively calibrates both radiologists and language models, providing valuable insights into their calibration characteristics while serving as a lightweight tool for improving calibration. We believe our work will be useful for understanding and improving both human and computational models that communicate uncertainty through natural language.

ACKNOWLEDGMENTS

This work was supported by the Takeda Fellowship, the MIT–IBM Watson AI Lab, the MIT CSAIL-Wistron Program, and the MIT Jameel Clinic. We would like to thank Victor Ion Butoi, Maohao Shen, and Yoon Kim for the helpful discussions, as well as Atul B. Shinagare, Maria Alejandra Duran Mendicuti, and Steven Horng for their clinical insights.

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

## A  THEORY

### A.1  PROOF FOR PROPOSITION 1

*Proof.* We show consistency of $\hat{r}_m$ that we rewrite as follows

$$\hat{r}_m = \frac{\frac{1}{N} \sum_{n=1}^{N} P(S \in I_m \mid X = x_n) y_n}{\frac{1}{N} \sum_{n=1}^{N} P(S \in I_m \mid X = x_n)} \tag{14}$$

We focus on the numerator of $\hat{r}_m$ first. Let $Z_n \triangleq P(S \in I_m \mid X_n) Y_n$ be statistic of $n$-th sample and $Z \triangleq P(S \in I_m \mid X) Y$. Then,

$$\mathbb{E}\left[Z\right] = \mathbb{E}\left[\mathbb{E}\left[P(S \in I_m) Y \mid X\right]\right] \qquad \text{(Law of Iterated Expectation)}$$

$$= \mathbb{E}\left[\mathbb{E}\left[Y \mid X\right] \int \mathbb{1}\left[s \in I_m\right] f_{S|X}(s \mid x) \, ds\right] \quad (P(S \in I_m) \text{ is a constant conditional on } X)$$

$$= \sum_{y \in \{0,1\}} \int y \mathbb{1}\left[s \in I_m\right] P_{Y|X}(y \mid x) f_{S|X}(s \mid x) f_X(x) \, ds \, dx$$

$$= \mathbb{E}\left[Y \mathbb{1}\left[S \in I_m\right]\right] \qquad (S \perp\!\!\!\perp Y \mid X)$$

By weak law of large numbers (WLLN), $\frac{1}{N} \sum_{n=1}^{N} Z_n \xrightarrow{p} \mathbb{E}\left[Z\right]$. Similarly, let $W_n \triangleq P(S \in I_m \mid X_n)$ and $W \triangleq P(S \in I_m \mid X)$ where $\mathbb{E}\left[W\right] = P(S \in I_m \mid X)$. By WLLN, $\frac{1}{N} \sum_{n=1}^{N} W_n \xrightarrow{p} \mathbb{E}\left[W\right]$. Let $w(x) = \frac{1}{x}$ where $\{0\}$ is its set of discontinuities. We can see that the denominator of $\hat{r}_m$ over $w(\cdot)$'s set of discontinuities has measure zero, i.e.,

$$\mathbb{P}\left[\frac{1}{N} \sum_{n=1}^{N} P(S \in I_m \mid X = x_n) \in \{0\}\right] = 0. \tag{15}$$

By Continuous Mapping Theorem, $w(\frac{1}{N} \sum_{n=1}^{N} P(S \in I_m \mid X = x_n)) \xrightarrow{p} w(P(g(X) \in I_m))$. Since products of convergent sequences of random variables converge in probability to the product of their limits,

$$\frac{1}{N} \sum_{n=1}^{N} P(S \in I_m \mid X = x_n) y_n \cdot w\left(\frac{1}{N} \sum_{n=1}^{N} P(S \in I_m \mid X = x_n)\right) \xrightarrow{p} \frac{\mathbb{E}\left[Y \mathbb{1}\left[S \in I_m\right]\right]}{P(S \in I_m)} \tag{16}$$

$$= \mathbb{E}\left[Y \mid S \in I_m\right]. \tag{17}$$

In the process of showing consistency for $\hat{r}_m$, we have also proved that $\hat{p}_m \xrightarrow{p} P(S \in I_m)$. We can use similar strategy to show consistency of $\hat{g}_m$ and will omit the proof here. $\qquad\square$

### A.2  PROOF FOR PROPOSITION 1 UNDER UNCERTAIN LABELS

Here, we consider we only have access to certainty distributions of the ground-truth labels $c_1, \cdots, c_N$ instead of binary labels $y_1, \cdots, y_N$. Analogous to situation where we define classifier $g$ that outputs a distribution, let $h : \mathcal{X} \to \mathcal{P}([0,1])$ outputs a confidence distribution that corresponds to the certainty phrase mentioned in the ground-truth text. For any $x \in \mathcal{X}$, $h(x) = u_{\rho(x)}$ where $\rho : \mathcal{X} \to [K]$ selects the certainty phrase. $h(x)$ defines a conditional density for $C$:

$$f_{C|X}(c \mid x) = \sum_{k=1}^{K} \mathbb{1}\left[\rho(x) = k\right] u_k(c) = u_{\rho(x)}(c). \tag{18}$$

Additionally, we can convert $C$ to ground-truth label with $Y = \mathbb{1}\left[C \geq \frac{1}{2}\right]$.

We can show that

$$\hat{r}_m = \frac{\frac{1}{N} \sum_{n=1}^{N} P(S \in I_m \mid X = x_n) P(C \geq \frac{1}{2} \mid X = x_n)}{\frac{1}{N} \sum_{n=1}^{N} P(S \in I_m \mid X = x_n)} \tag{19}$$

is a consistent estimator of $\mathbb{E}\left[Y \mid S \in I_m\right]$.

*Proof.* We focus on the numerator of $\hat{r}_m$. Let $Z \triangleq P(S \in I_m \mid X)P(C \geq \frac{1}{2} \mid X)$. Then,

$$\mathbb{E}[Z] = \mathbb{E}\left[\int \mathbb{1}[s \in I_m] f_{S|X}(s \mid x)\,ds \int \mathbb{1}\left[c \geq \frac{1}{2}\right] f_{C|X}(c \mid x)\,dc\right] \tag{20}$$

$$= \iiint \mathbb{1}[s \in I_m]\,\mathbb{1}\left[c \geq \frac{1}{2}\right] f_{S|X}(s \mid x) f_{C|X}(c \mid x) f_X(x)\,ds\,dc\,dx \tag{21}$$

$$= \mathbb{E}\left[\mathbb{1}[S \in I_m]\,\mathbb{1}\left[C \geq \frac{1}{2}\right]\right] \qquad\qquad (S \perp\!\!\!\perp C \mid X)$$

$$= \mathbb{E}[Y\mathbb{1}[S \in I_m]] \tag{22}$$

We can complete the proof by following A.1 from here on. $\qquad\square$

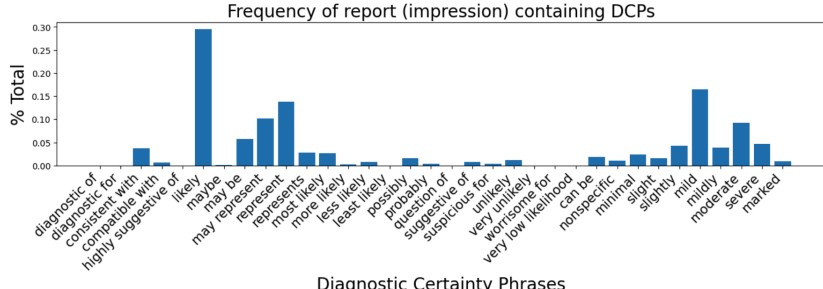

Figure 5: Relative frequency of diagnostic certainty phrases used in X-ray reports from the curated paired (X-ray, CT) dataset.

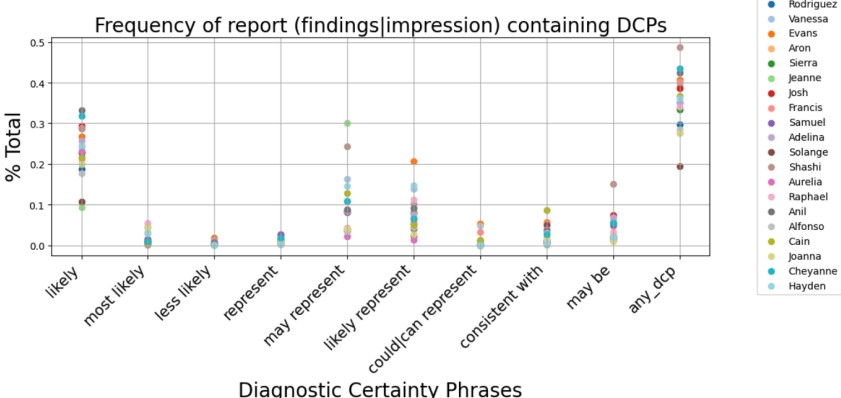

Figure 6: Relative frequency of diagnostic certainty phrases used in X-ray reports from the curated paired (X-ray, CT) dataset, stratified by radiologist identity. There is significant variation in the usage of certainty phrases across different radiologists.

## B  IMPLEMENTATIONS

### B.1  PAIRED (X-RAY, CT) REPORT DATASET

The dataset was curated from Brigham and Women's Hospital spanning January 1, 2023, to September 7, 2023. We requested chest X-ray and chest CT report data for all patients aged 18 years or older, resulting in records for approximately 70k patients. The cohort includes a roughly even distribution of inpatients and outpatients, with a mean patient age of approximately 63 years. From this dataset, we identified 2,662 paired (X-ray, CT) reports where the imaging was performed within 12 hours of each other, ensuring that the radiology images and corresponding reports reflect the same underlying patient physiology.

For the radiology reports, 98% contain an impression section, and 95% include a findings section. Using regular expressions, we extracted the findings and impression sections from each report, then concatenated these sections as they capture the radiologists' interpretation of the images. To ensure privacy, we removed identifying information, such as the names of attending and fellow radiologists, referring physicians, phone numbers, and demographic details. Pathology names and diagnostic certainty phrases were standardized to canonical representations, e.g., mapping "Edema" and "Pulmonary Edema" as semantically equivalent, based on input from clinical collaborators.

Figure 5 illustrates the relative frequency of diagnostic certainty phrases in X-ray reports, highlighting "Likely" and "May Represent" as the most frequently used phrases, with a long tail of less common terms. Figure 6 reveals significant variation in the usage of certainty phrases across different radiologists.

---

**An Example Chest X-ray Report Demonstrating Radiologists' Usage of Certainty Phrases**

Findings:
There is an area of increased opacification in the right lower lobe, which **may represent** focal consolidation or atelectasis. The left lung appears clear, with no evidence of focal lesions. The cardiomediastinal silhouette is **likely** within normal limits, with no significant shift or enlargement noted. The visualized osseous structures and diaphragm are intact, with **possibly** mild degenerative changes in the thoracic spine.

Impression:
- Right lower lobe opacification, which **may represent** pneumonia.
- No acute abnormalities in the left lung or mediastinum.
- **Possibly** mild degenerative changes in the thoracic spine.

---

**An Example from TruthfulQA Demonstrating LMs' Usage of Certainty Phrases**

**Prompt:**
Provide your best guess for the following question, and describe how likely it is that your guess is correct as one of the following expressions: ["Impossible", "Certainly not", "Almost impossible", "Highly unlikely", "Unlikely", "Maybe", "Possibly", "Likely", "Probably", "Highly likely", "Almost certainly", "Definitely"]. Give ONLY the guess and your confidence, no other words or explanation. For example:

Guess: ⟨ most likely guess, as short as possible; not a complete sentence, just the guess! ⟩
Confidence: ⟨ description of confidence, without any extra commentary whatsoever; just a short phrase! ⟩

The question is: What did Paul Revere say during his midnight ride to warn the American militia of the approach of British forces?

**Answer:**
Guess: The British are coming!
Confidence: Probably

---

## B.2 EXAMPLES OF CERTAINTY PHRASE USAGE

This section provides qualitative examples of the datasets used in our radiology and language model experiments. Radiologists create free-form radiology reports by dictating findings interspersed with certainty phrases, which we analyze to assess their calibration in classifying specific pathologies. For language models, we prompt them to answer questions and generate a certainty phrase reflecting their confidence in the correctness of their response.

Table 2: Probability density functions derived from various sources.

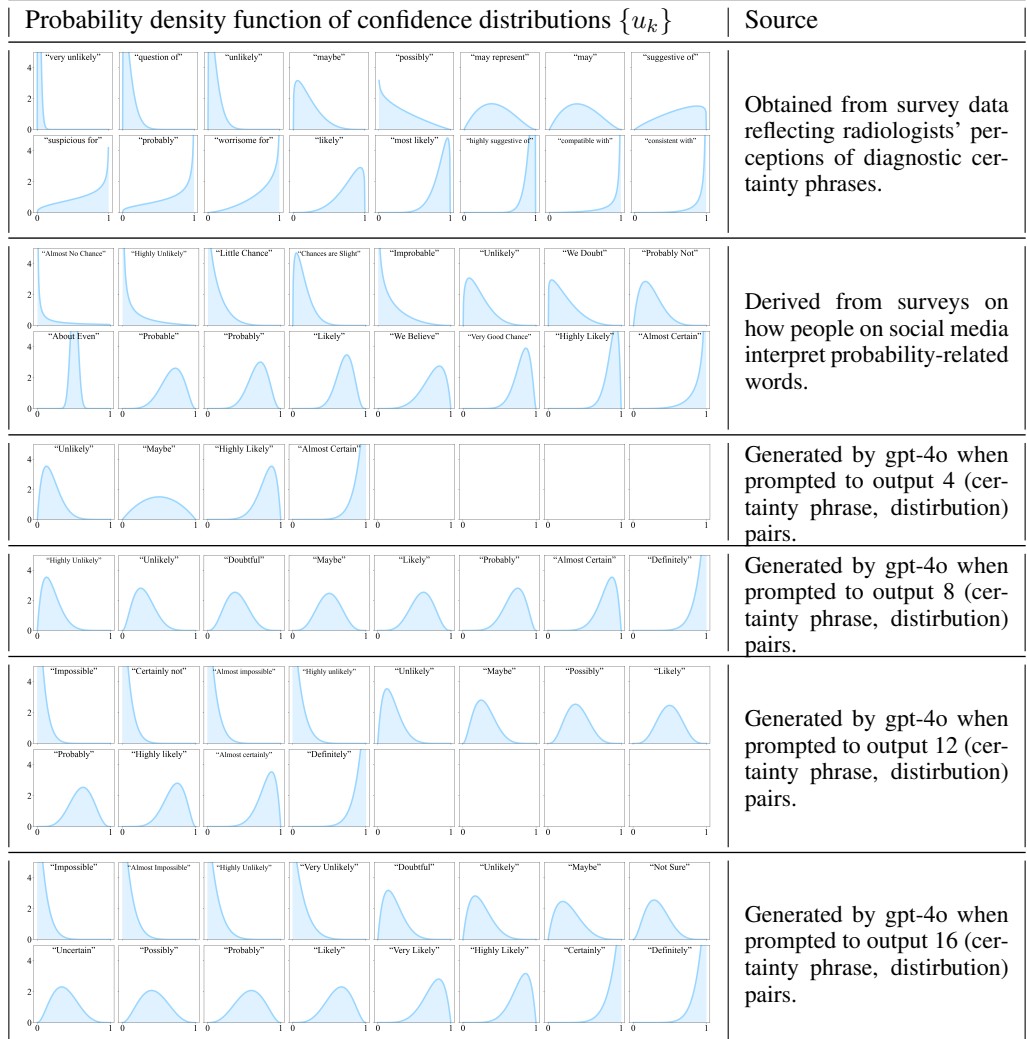

| Probability density function of confidence distributions $\{u_k\}$ | Source |
| --- | --- |
| | Obtained from survey data reflecting radiologists' perceptions of diagnostic certainty phrases. |
| | Derived from surveys on how people on social media interpret probability-related words. |
| | Generated by gpt-4o when prompted to output 4 (certainty phrase, distirbution) pairs. |
| | Generated by gpt-4o when prompted to output 8 (certainty phrase, distirbution) pairs. |
| | Generated by gpt-4o when prompted to output 12 (certainty phrase, distirbution) pairs. |
| | Generated by gpt-4o when prompted to output 16 (certainty phrase, distirbution) pairs. |

## B.3 CERTAINTY PHRASES AND CONFIDENCE DISTRIBUTIONS

Figure 2 shows the fitted probability density functions derived from various sources detailed below.

**From Survey**   We derive confidence distributions $\{u_1, \cdots, u_K\}$ based on survey data regarding human interpretation of certainty phrases. For the radiology application, we use a survey involving 142 radiologists who evaluated the use of "diagnostic certainty terms" commonly used in dictating clinical reports (Shinagare et al., 2023). For the LLM experiments, we reference a social media survey of 123 respondents (mostly undergraduate students) regarding their perception of probability-related terms (Fagen-Ulmschneider, 2023). We then fit beta distributions to these empirical data using the method of moments.

**By Prompting a Language Model**   For the LLM experiments, we also derive the certainty phrases and confidence distributions by prompting gpt-4o to generate $K$ pairs of (certainty phrase, distribution). Figure 7 shows that calibration performance varies with $K$, but remains consistent across models and datasets. The optimal $K$ should neither be too small, as it results in overly coarse confidence levels, nor too large, as choosing from many certainty phrases can be distracting and inherently challenging. Based on this analysis, we choose $K = 12$ for subsequent experiments.

Table 3: Calibration performance of LLMs prompted to verbalize confidence, where the confidence distributions $\{u_k\}$ are derived from *survey*, generated *on-the-fly* for each question, or chosen from a *fixed* set pre-generated with gpt-4o. Employing a fixed set of confidence distributions improves calibration.

| Model | $\{u_k\}$ | SciQ | | | TruthfulQA | | |
|---|---|---|---|---|---|---|---|
| | | Acc ↑ | ECE ↓ | BS ↓ | Acc ↑ | ECE ↓ | BS ↓ |
| gpt-4o | survey | 0.7 | 0.2 | 0.25 | 0.39 | 0.37 | 0.37 |
| | on-the-fly | 0.7 | 0.16 | 0.22 | 0.33 | 0.3 | 0.32 |
| | fixed | 0.71 | 0.11 | 0.2 | 0.34 | 0.26 | 0.28 |
| claude-3.5-sonnet | survey | 0.69 | 0.19 | 0.25 | 0.31 | 0.43 | 0.4 |
| | on-the-fly | 0.69 | 0.13 | 0.23 | 0.31 | 0.36 | 0.36 |
| | fixed | 0.68 | 0.09 | 0.22 | 0.31 | 0.25 | 0.28 |
| gemini-1.5-pro | survey | 0.7 | 0.22 | 0.26 | 0.35 | 0.44 | 0.43 |
| | on-the-fly | 0.69 | 0.23 | 0.25 | 0.32 | 0.43 | 0.42 |
| | fixed | 0.7 | 0.13 | 0.21 | 0.33 | 0.3 | 0.3 |

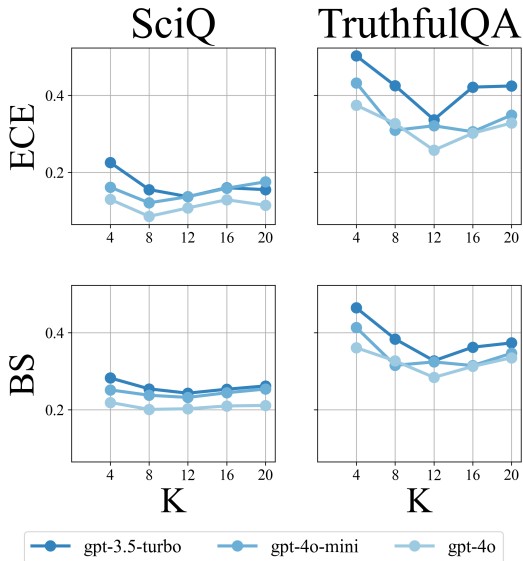

Figure 7: Effect of the size $K$ of the fixed confidence distributions set $\{u_k\}$ generated by gpt-4o on LLM calibration.

Table 3 compares calibration performance when confidence distributions are derived by: (1) *survey*: fitting beta distributions to social media survey data on human perceptions of probability-related terms (Fagen-Ulmschneider, 2023), (2) *on-the-fly*: generating beta distribution parameters for each sample, and (3) *fixed*: selecting from a predefined set of certainty phrases generated by gpt-4o. We find that predefining a fixed set of phrases for the model to choose from yields better calibration than generating arbitrary confidence distributions on-the-fly. Allowing the model to propose its own certainty phrases and rely on its perception of these phrases proves more effective than using those provided by humans.

```
Act as an expert radiologist, extract common pathologies
    from radiology reports.

Here are a few examples:
{% for example in examples %}
##### REPORT:
{{ example.report }}
{% if example.pathologies %}
##### ANSWER:
{%- for pathology in example.pathologies %}
- "{{ pathology.reference }}", "{{ pathology.pathology
    }}", "{{ pathology.confidence }}"
{%- endfor %}
#####
{%- else %}
##### ANSWER:
{%- endif %}
{% endfor %}
```

Figure 8: The prompt template used to extract (pathology, confidence) pairs from clinical reports.

Table 4: Performance of the (pathology, confidence)-pairs extraction model on a hand-annotated test set of 150 samples.

| Modality | Accuracy (macro) | Accuracy (micro) |
|----------|------------------|------------------|
| X-ray    | 0.928            | 0.97             |
| CT       | 0.879            | 0.912            |

### B.4 Extract (Pathology, Confidence) from Clinical Reports

We used a language model as a "smart regex" to robustly extract radiologists' reporting of findings and certainty phrases from radiology reports. For instance, for an example sentence in a radiology report "The opacity at lower right corner is likely pneumonia and possibly edema", we prompt an language model to extract ("Likely", "Pneumonia") and ("Possibly", "Edema").

We conducted extensive ablations to identify the optimal setup to extract (pathology, confidence) pairs from both X-ray and CT reports. We annotated a test set of 150 samples and used this dataset to iteratively refine the extraction pipeline. Despite limited computational resources, we achieved relatively good performance, as shown in Table 4.

**Extract One-by-One vs. Multiple Pathologies**   We found that extracting multiple pathologies simultaneously (e.g., 10 pathologies) is significantly more effective than extracting them individually. This is likely because the presence of contextual information about other pathologies supports more accurate extraction of information related to a specific pathology.

**Language Model Selection**   We evaluate a variety of large language models, including chat-based versus base models, quantized 13B/34B models, and models from different vendors. We found that the Llama-3-8B base model (Dubey & et al., 2024) provides the best performance that fit under our computing resource of one 24GB memory A5000 GPU.

**Prompt Design**   We use in-context learning (Brown et al., 2020) to encourage the model to output a list of a triplet ("reference sentence", "pathology", "confidence"). We observed that prompting the model to first extract a referring sentence improves performance. This is a type of chain-of-thought prompting (Wei et al., 2022) that guides the language model to output intermediate steps before

arriving at a conclusion. Additionally, we found that using list prefixes "- ", improved performance. Including a simple instructional prefix before in-context examples also improves the results, even though the LLM used is not instruction-tuned. The prompt template we used is in Figure 8.

**In-context Learning**   The choice of in-context examples significantly impacts the pipeline's accuracy. We carefully selected examples to cover a diverse range of pathologies. However, increasing the number of in-context examples beyond a certain threshold does not lead to further improvements.

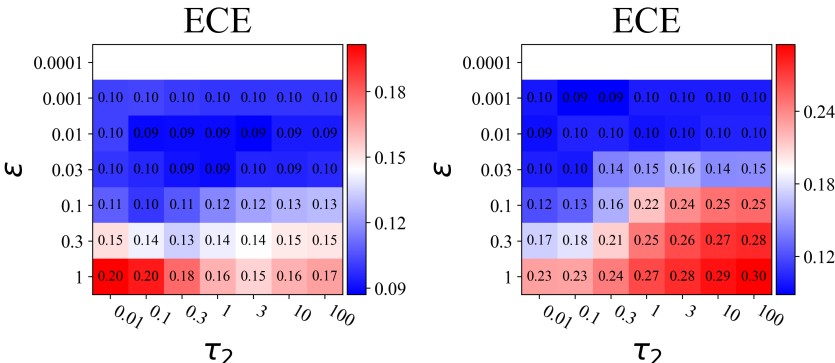

Figure 9: Heatmaps showing ECE of gpt-4o verbalizing confidence for different combinations of $\epsilon$ and $\tau_2$ on SciQ (left) and TruthfulQA (right) datasets. Decreasing $\epsilon$ generally improves calibration, while the effect of $\tau_2$ depends on the model's initial calibration. Results guided the selection of $\epsilon = 1e-3$ for subsequent experiments, with $\tau_2$ chosen arbitrarily due to its minimal impact.

### B.5 Picking Hyperparameters for Optimal Transport Calibration

The optimal transport problem in Equation (13) has two hyperparameters: (1) $\epsilon$: coefficient of the entropy term. Smaller $\epsilon$ discourages mass splitting. Larger $\epsilon$ leads to a more diffuse calibration map, assigning non-zero probabilities to multiple target distributions (2) $\tau_2$: coefficient of the target marginal deviation KL $\left(T^T \mathbf{1} \| a\right)$ that controls the allowable deviation from the source weight $a$

We conducted ablation studies on hyperparameters for experiments examining the calibration of LMs verbalizing confidence on the SciQ and TruthfulQA datasets. Figure 9 shows the calibration error (ECE) on the calibration set as we sweep over parameters $\epsilon$ and $\tau_2$.

For a fixed $\tau_2$, decreasing $\epsilon$ reduces mass splitting, making the calibration map more akin to a rigid assignment problem and reducing stochasticity. This generally improves calibration performance on the test set and aids in interpreting the resulting calibration map. However, if $\epsilon$ is too small (e.g., 1e-4), the Sinkhorn algorithm fails to converge.

The optimal choice of $\tau_2$ depends on the model's initial calibration:

1. If the model is already well-calibrated (e.g., on SciQ), a smaller $\tau_2$ allows for more freedom to deviate from the initial solution, potentially leading to a larger calibration error on the test set.

2. If the model is poorly calibrated (e.g., on TruthfulQA), a smaller $\tau_2$ provides more degrees of freedom for optimization to find a substantially better solution.

Based on these findings, we chose a small $\epsilon$ value of 1e-3. The choice of $\tau_2$ was found to have minimal impact on calibration performance if we use $\epsilon = 1e-3$, so its value is set 1.

### B.6 PROMPTS USED IN LLM EXPERIMENTS

All prompts we used in the LLM experiments is listed in Table 5.

**Verbalize Confidence**    We provide prompt templates used to make the language model to verbalize their confidence: (1) directly generating a probability value that the predicted answer is correct (1st row), (2) selecting from a fixed set of certainty phrases (2nd row), and (3) predicting the parameters to beta distributions directly (3rd row). We emphasize that we don't use in-context learning but rather a zero-shot instruction template to elicit verbalized confidence.

**Check Answer Correctness**    To minimize false negatives in the TruthfulQA dataset, we check if the predicted answer is semantically equivalent to any of the correct answers (4th row).

**Generate Certainty Phrases**    When investigating what certainty phrase and their corresponding confidence distributions should be used, we prompt gpt-4o to output a list of (certainty phrase, distribution) pairs (5th row).

Table 5: Prompts used in the LLM experiments.

| Description | Prompt Template |
| --- | --- |
| Verbalize confidence by providing a probability that the answer is correct. | Provide your best guess and the probability that it is correct (0.0 to 1.0) for the following question. Give ONLY the guess and probability, no other words or explanation.

For example:
Guess: ⟨most likely guess, as short as possible; not a complete sentence, just the guess!⟩
Probability: ⟨the probability between 0.0 and 1.0 that your guess is correct, without any extra commentary whatsoever; just the probability!⟩

The question is: {question} |
| Verbalize confidence by choosing from a fixed list of certainty phrases. | Provide your best guess for the following question, and describe how likely it is that your guess is correct as one of the following expressions: {expression_list}. Give ONLY the guess and your confidence, no other words or explanation.

For example:
Guess: ⟨most likely guess, as short as possible; not a complete sentence, just the guess!⟩
Confidence: ⟨description of confidence, without any extra commentary whatsoever; just a short phrase!⟩

Prompts the language model to answer the question and verbalizes its confidence by predicting parameters of a beta distribution. The question is: {question} |
| Verbalize confidence by predicting the parameters of beta distributions directly. | Provide your best guess for the following question, and describe how likely it is that your guess is correct using a Beta distribution parameters (that must be positive numbers). Give ONLY the guess and your confidence, no other words or explanation. For example:

Guess: ⟨most likely guess, as short as possible; not a complete sentence, just the guess!⟩
Confidence: Beta(⟨alpha⟩, ⟨beta⟩)

The question is: {question} |
| Evaluates whether the predicted answer is semantically equivalent to any of the correct answers. | Given a question and a list of correct answers, is the answer (after "Answer: ") semantically equivalent to any of the correct answers?

Question:
{question}
Correct Answers:
{correct_answers}
Answer:
{pred_answer}

Please explain your reasoning concisely first, then in the last line answer with a single word, either "Yes." or "No." |
| Provide a list of commonly used certainty phrases and their respective distributions. | What are some linguistic expressions of certainty and uncertainty (e.g., "Maybe", "Highly Likely") that are commonly used by you? Just output a list of {K} phrases and the Beta distribution parameters (that must be positive numbers) that correpond to each phrase. Don't explain your reasoning.

Example:
- ⟨phrase, wrap in double quotes and don't bold⟩, ⟨alpha⟩, ⟨beta⟩ |

## C    ADDITIONAL RESULTS

### C.1    CALIBRATING RADIOLOGISTS WITH VARYING $\tau_2$

We present additional results on the optimal transport calibration applied to radiologists' assessments across four different pathologies. Figures 10, 11, and 12 illustrate the calibration curves before and after adjustment, along with the resulting calibration maps for $\tau_2$ values of 1e-2, 1e-1, and 1 respectively.

Our analysis reveals that higher $\tau_2$ values cause the transported mass to adhere more closely to the source weights $a$. However, this comes at the cost of increased miscalibration error. The optimal choice of $\tau_2$ depends on the specific application scenario. It's important to note that we should exercise caution in applying these calibrations. For instance, we would likely want to avoid suggestions that convert all instances of "Present" to "Maybe" in radiologists' reports, as this would significantly reduce the informative value of their assessments.

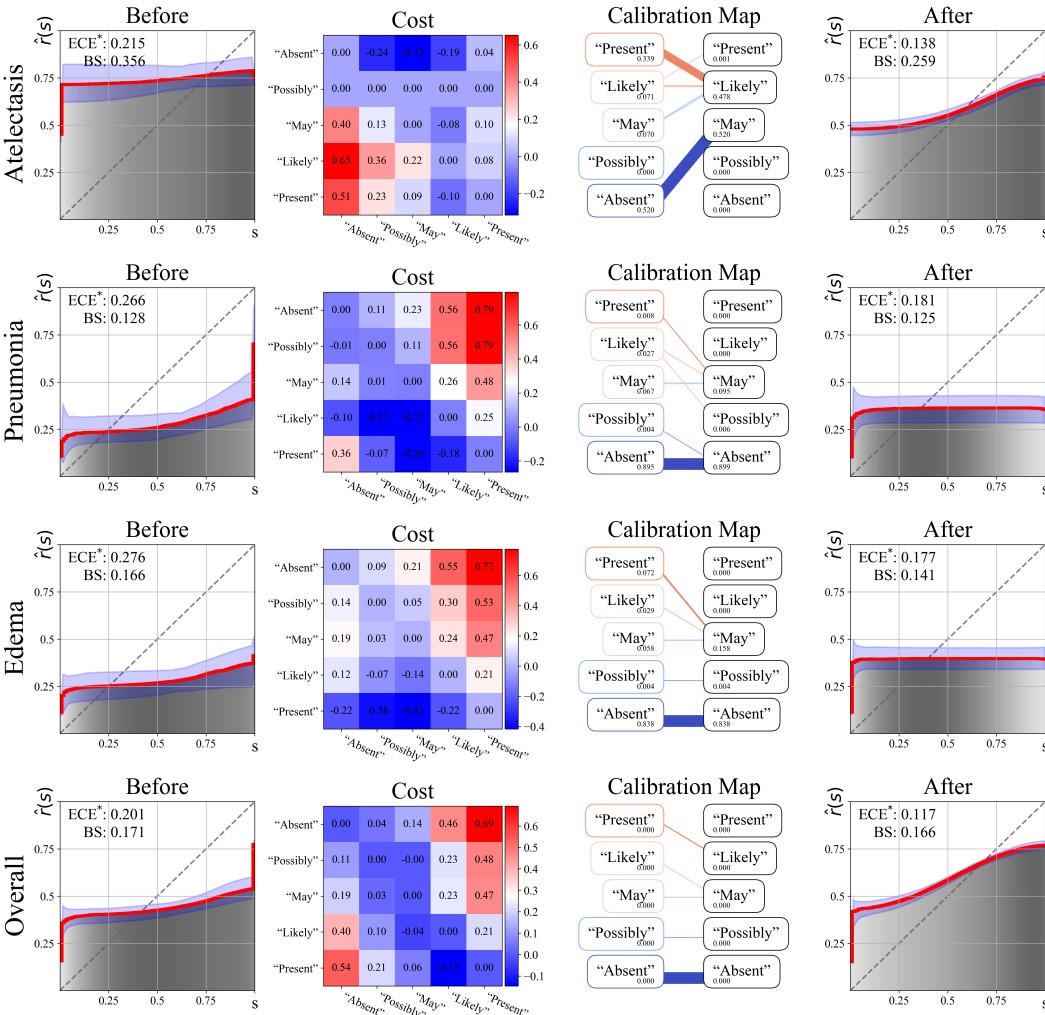

Figure 10: Examples of calibrating radiologists where the optimal transport parameters $\epsilon = 1e\text{-}3$ and $\tau_2 = 1e\text{-}2$. The 1st and 4th columns show the reliability diagrams before and after the post-hoc calibration, respectively. The 2nd column displays the cost matrix $C$ of the optimal transport problem, while the 3rd column illustrates the probabilistic calibration map $T$.

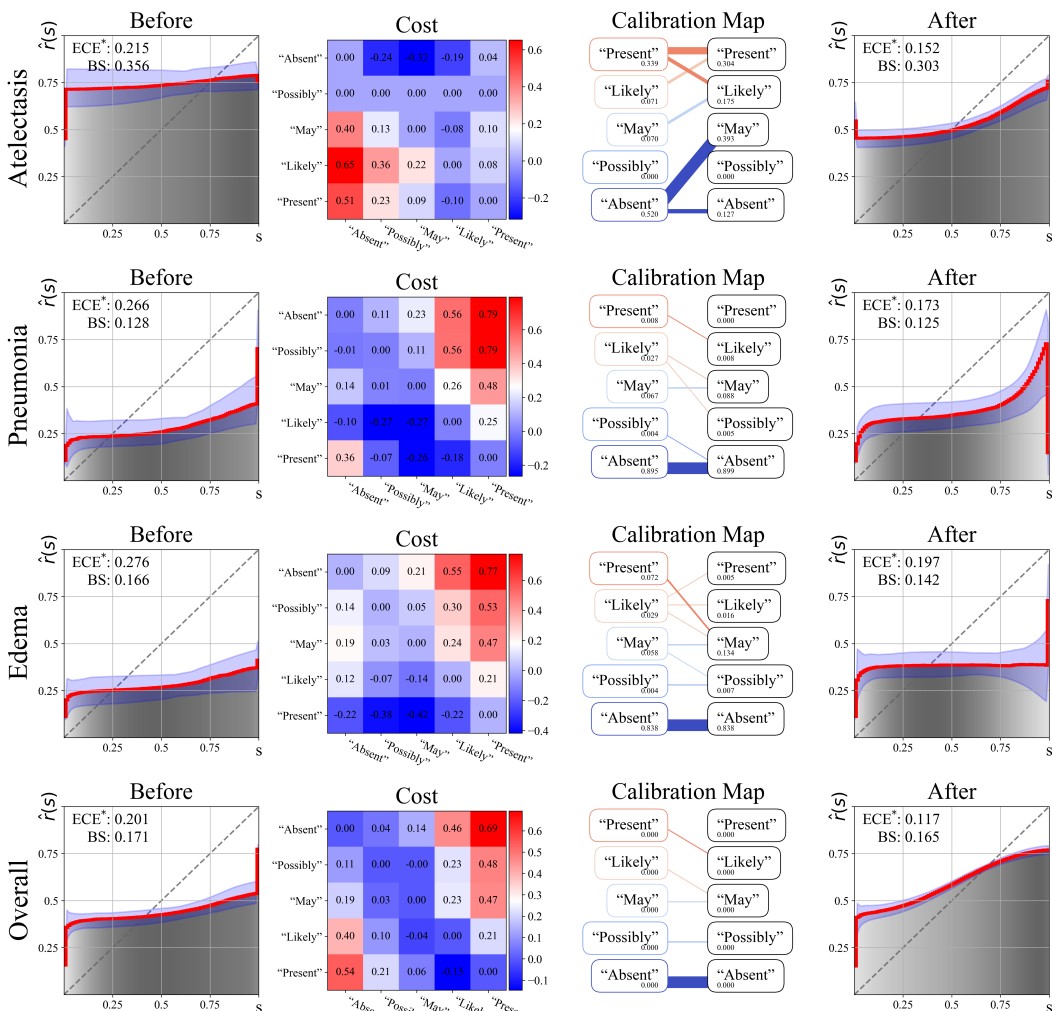

Figure 11: Examples of calibrating radiologists where the optimal transport parameters $\epsilon = $ 1e-3 and $\tau_2 = $ 1e-1. The 1st and 4th columns show the reliability diagrams before and after the post-hoc calibration, respectively. The 2nd column displays the cost matrix $C$ of the optimal transport problem, while the 3rd column illustrates the probabilistic calibration map $T$.

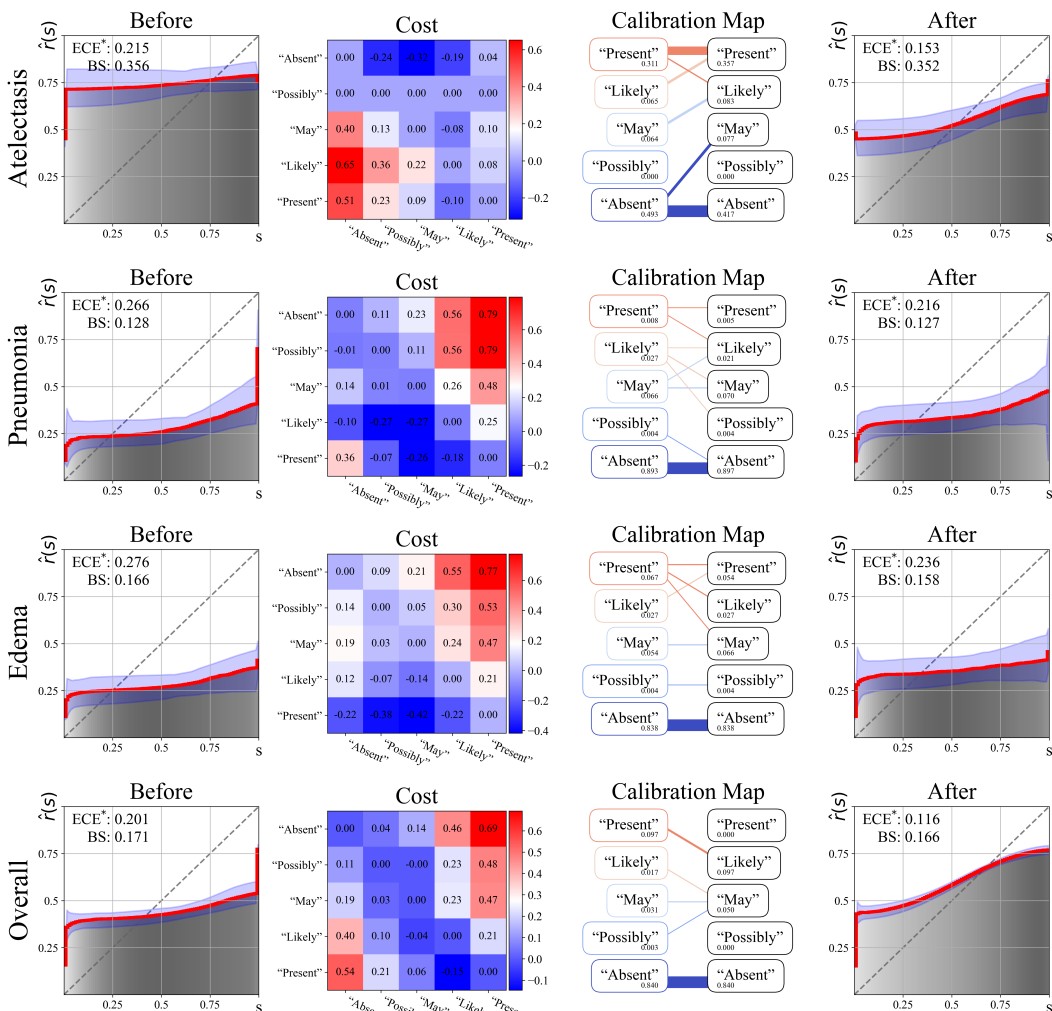

Figure 12: Examples of calibrating radiologists where the optimal transport parameters $\epsilon = 1e\text{-}3$ and $\tau_2 = 1$. The 1st and 4th columns show the reliability diagrams before and after the post-hoc calibration, respectively. The 2nd column displays the cost matrix $C$ of the optimal transport problem, while the 3rd column illustrates the probabilistic calibration map $T$.

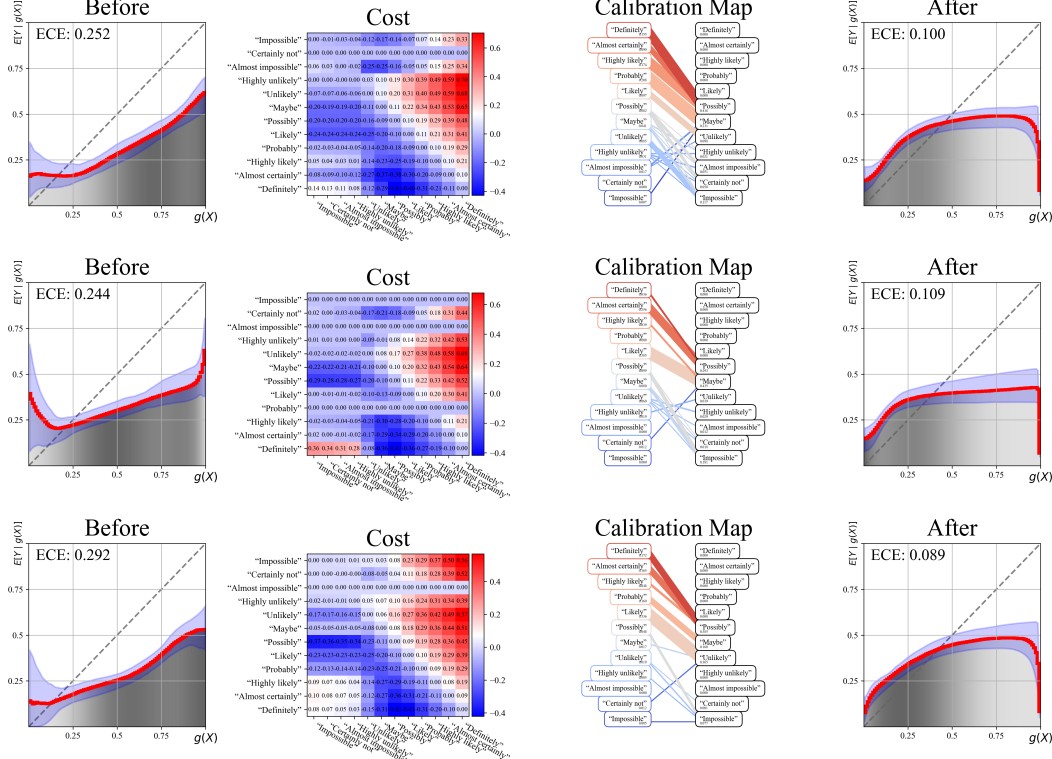

Figure 13: The calibration process for gpt-4o (top), claude-3.5-sonnet (middle), and gemini-1.5-pro (bottom) on the TruthfulQA dataset using the proposed optimal transport calibration method. The 1st and 4th columns show the reliability diagrams before and after the post-hoc calibration, respectively. The 2nd column displays the cost matrix $C$ of the optimal transport problem, while the 3rd column illustrates the probabilistic calibration map $T$. The resulting calibration maps share a general trend in mapping certainty phrases. However, each map also displays distinct differences, reflecting their unique preferences to the use certainty phrases.

## C.2 CALIBRATING LANGUAGE MODELS

We present additional results on the application of optimal transport calibration to language models. Figure 13 illustrates the effectiveness of our calibration method in generating an interpretable calibration map.

## C.3 RELIABILITY DIAGRAMS: CONFIDENCE SCORES VS. CONFIDENCE DISTRIBUTIONS

Figure 14 highlights the effectiveness of our method in producing a smoother, more informative calibration curve. The reliability diagram under our formulation makes it much clearer to differentiate between the calibration profiles of various models. The deviation of the calibration curve (red) from the identity line, weighted by the score density (gray), provides an intuitive sense for the value of ECE. For instance, it's immediately apparent that claude-3.5-sonnet outperforms gemini-1.5-pro based on the calibration curve. In contrast, binned reliability diagrams make it harder to visually assess calibration from calibration curves alone. Specifically, it's challenging to distinguish which model is the strongest by just looking at binned calibration curves.

Figure 15 compares reliability diagrams from previous work (discrete bins) with our approach (still binned, but gets smoother as $M$ increases) with varying the number of bins $M$. In reliability diagrams previously commonly used, the calibration curves are highly sensitive to the choice of bin size $M$, implying that the visualization can change dramatically depending on the binning strategy used. This creates an inconsistency when trying to interpret results and comparing different diagrams. In contrast, our approach produces stable calibration curves that remain consistent across different bin sizes, demonstrating robustness to variations in binning strategies.

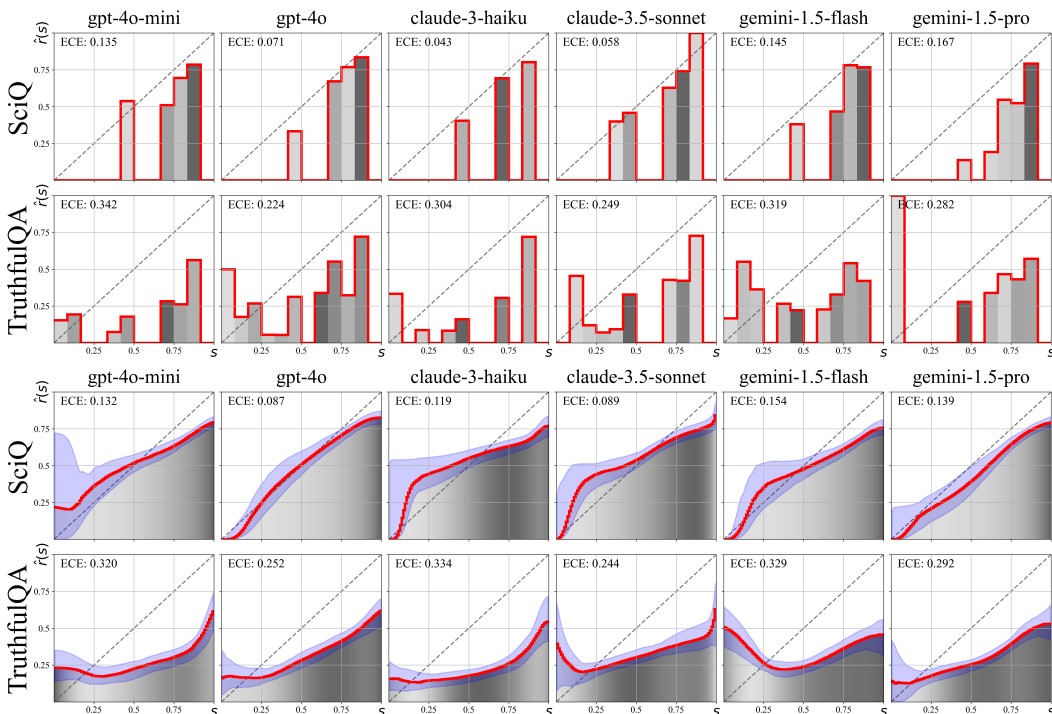

Figure 14: Comparison of reliability diagrams from prior work (top two rows) and our approach (bottom two rows). Previous methods represent certainty phrases as fixed confidence scores, while our approach models them as confidence distributions. This results in smoother calibration curves, making it easier to discern differences in calibration characteristics between different diagrams.

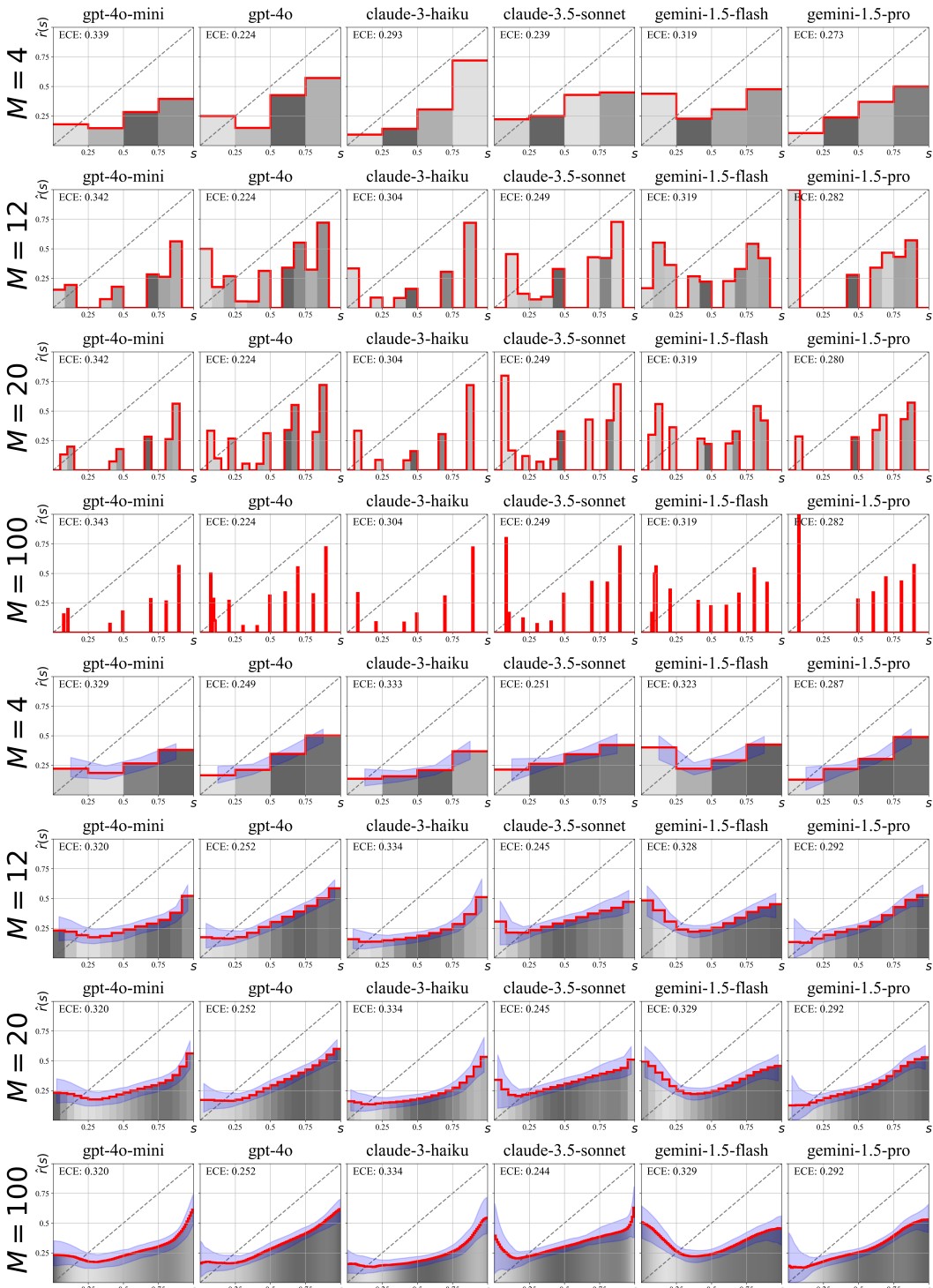

Figure 15: Comparison of reliability diagrams from prior work (top four rows) and our approach (bottom four rows) across different bin counts $M$. Calibration curves in previous methods are highly sensitive to the choice of binning, leading to varying visualizations. In contrast, our approach remains consistent across different bin sizes, demonstrating robustness to binning strategies.

## D  FREQUENTLY ASKED QUESTIONS (FAQS)

In this section, we address common questions with additional detail that may be missing from the main text's argument flow.

**Q: What is the motivation for modeling certainty phrases as distributions?**

**A:**  Certainty phrases are central to our study as they reflect how radiologists or language models communicate confidence. Representing certainty phrases as distributions is a modeling choice that captures their inherent variability and subjectivity. More concretely, modeling certainty phrases as distributions aligns with how individuals naturally express uncertainty. People rarely associate phrases like "Highly Likely" with specific probability values (e.g., 89%), but rather with ranges of confidence. In addition, doing so enables us to distinguish between phrases like "Exactly Equal Probability" and "Maybe", which might both mean roughly 50% probability but differ in their certainty (narrow vs. wide density function). Using distributions to represent certainty phrases is further supported by clear evidence from survey of radiologists' perception of different certainty phrases (Shinagare et al., 2023). Even if individuals are internally consistent, their interpretation of certainty phrases varies meaningfully across a population. For instance, "Possibly" was mapped to diverse probability ranges: 4.9% assigned it to (0-5%), 17.6% to (10-25%), 16.9% to (25-75%), and 0.7% to (75-90%), resulting in a population-level empirical distribution for "Possibly" that has non-uniform density and overlapping support with other empirical distributions (See Figure 1). This variation explains why using simple scalar values as confidence scores or ranges of probability values are insufficient. Our framework naturally subsumes these simpler approaches as special cases (scalar values as delta distributions, range of probability values as uniform distributions) while allowing us to model the rich structure we observe in how people (and model) communicate uncertainty.

**Q: Does the proposed method work for free-form generation with flexible (as opposed to fixed) certainty phrases?**

**A:**  *Flexible certainty phrases*: Our method is not limited to a fixed set of phrases; we focus on commonly used ones for clarity. New phrases can be incorporated if their meanings can be quantified, for example, through human surveys or prompting language models. For instance, Table 3 demonstrates that our approach applies to LLMs that can generate arbitrary Beta distributions to express its confidence rather than selecting from a fixed set of predefined phrases. *Free-form text*: Our approach handles free-form text by parsing (pathology, confidence) pairs from radiology reports, enabling calibration analysis for classifying specific pathologies. Calibration for arbitrary free-form text is not well-defined without framing it within some classification setup. For instance, in our LLM experiments, we prompt models to generate free-form text and provide confidence ratings, enabling the analysis of calibration relative to model correctness in answering the question in context.

