# OpenReview forum: "Calibrating Expressions of Certainty"
_ICLR.cc/2025/Conference — ICLR 2025 Poster_

### Official Review · Reviewer_itYd · 2024-10-19

**Soundness:** 3
**Presentation:** 3
**Contribution:** 3
**Rating:** 6
**Confidence:** 3

**Summary:**

This paper describes a novel approach to calibrate certainty phrases directly instead of first mapping these phrases to an explicitly confidence score and next calibrating these scores. The method uses probability distributions from a data set with expressions and confidence scores to find a “translation” mapping. The model is tested on radiology reports with human descriptions and on Q&A tasks for LLMs. Their approach performs similar to the traditional methods that use indirect mappings. The main advantage is that their approach uses a direct mapping from certainty phrase to certainty phrase representing global value ranges.

**Strengths:**

- calibration method that does not require an explicit mapping from phrase to score
- intuitively models peoples use of uncertainty expressions to calibrate for populations
- strong mathematical modeling (as far as I can judge as a non-expert on this)

**Weaknesses:**

- the model does not outperform standard SOTA approaches
- the model seems to remove outlier expressions and calibrates to the middle area
- the mathematical modeling is dense and difficult to understand for non-experts
- misses a deeper analysis and clarifying examples about the semantics and use of uncertainty phrases
- introduction promises that it can help radiologists to report more consistently but cannot find and evidence of this in the paper.

**Questions:**

- the model does not outperform standard SOTA approaches
- the model seems to remove outlier expressions and calibrates to the middle area
- the mathematical modeling is dense and difficult to understand for non-experts
- misses a deeper analysis and clarifying examples about the semantics and use of uncertainty phrases
- introduction promises that it can help radiologists to report more consistently but cannot find and evidence of this in the paper.

---

> ### Author Response · Authors · 2024-11-19
> **response**
>
> We appreciate the reviewer’s thoughtful feedback and provide clarifications below:
>
> **Comparison to Baseline Calibration Methods**: We apologize that our description of the experimental results in Table 1 did not make it sufficiently clear that our goal is not to significantly improve calibration of numerical scores (which is what Table 1 evaluates) but rather show that even if one reduces the output natural language expressions of certainty to scalar values (by simply taking the mean of the confidence distribution), our method remains competitive and does not compromise performance in terms of accuracy. We emphasize that our method's key advantage isn't in outperforming baseline calibration methods, but in directly operating on and producing natural language certainty phrases - a property that existing methods fundamentally lack. This makes our approach uniquely suited for improving human calibration in real-world settings, as it provides actionable guidance (e.g., suggesting radiologists use "May" instead of "Present" in their reporting to mitigate overconfidence) rather than abstract probability adjustments that can't be easily understood by humans. The goal is to maintain strong calibration performance while enabling calibration in natural language settings where traditional calibration methods cannot be meaningfully applied.
>
> **Outlier Expression and Mapping to the Middle**: The observed mapping of certainty phrases to "middle" categories (e.g., "Maybe") in Figure 3 reflects dataset-specific characteristics, e.g., radiologists have high false negative and false positive rates, rather than a limitation of our calibration method. For instance, Appendix C.2 demonstrates a different calibration map that shifts the use of certainty phrases from high confidence to low confidence to reduce overconfidence, e.g., "Definitely" to "Possibly", ..., "Unlikely" to "Impossible".
>
> **Dense Math Derivation**: To ensure clarity, it’s necessary that we provide concrete mathematical derivations of our method. If the reviewer has specific questions about the derivation, we will gladly clarify and expand to improve readability.
>
> **Clarifying Examples & Deeper Analysis**: To address this, we have now included qualitative examples in Appendix B.2, illustrating how radiologists and language models use certainty phrases in the analyzed datasets.  We will gladly provide additional analyses and examples if the reviewer has specific suggestions in mind that will probe the utility of the proposed approach.
>
> **Improve Calibration of Radiologists**: Figure 3 demonstrates how radiologists' calibration improves by following our calibration method perfectly, with ECE reducing from 0.215 to 0.138 for atelectasis and from 0.276 to 0.177 for edema. We are conducting a follow-up clinical study, that is outside the scope of this paper, to examine how receptive radiologists are to calibration-improving suggestions provided by our method, e.g., use "May" or "Likely" instead "Present" in your reporting to address overconfidence.

---

### Official Review · Reviewer_qs5L · 2024-11-01

**Soundness:** 4
**Presentation:** 4
**Contribution:** 4
**Rating:** 8
**Confidence:** 3

**Summary:**

The paper measures and calibrate expressions of certainty made in natural language (e.g., likely, unlikely). By treating certainty phrases
as distributions over the probability simplex, the authors generalize existing estimators for miscalibration metrics, such as the expected calibration error. The work has real-world implications in fields like radiology.

**Strengths:**

- A good understanding and presentation of the existing literature on calibration.
- Using confidence distributions makes the proposed estimator more robust to increasing the number of bins.
- Propose a novel framing of calibration as a composition of source confidence distributions, an optimal transport map to target confidence distributions, and an indexing function.
- Provides potential real-world use cases, such as calibrating human expressions of uncertainty in medicine and LLM expressions of uncertainty.

**Weaknesses:**

- The paper focuses on binary classification, which could make its impact limited.
- Conversely, the relative complexity of the approach to non-statisticians could limit its adoption.
- The candidate confidence distributions could in practice be very large, which could make some of the optimization problems very slow to solve.

**Questions:**

In practice, if humans are already mentally correcting for the under/over-estimation of expressed uncertainty (either by a doctor or LLM), couldn't such calibration lead to worse outcomes, since their mental models no longer apply while the calibrated expressions are still not perfect? This is more of a human problem than a technical one, but just a thought.

---

> ### Author Response · Authors · 2024-11-19
> **response**
>
> We appreciate the reviewer’s thoughtful feedback and provide clarifications below:
>
> **Focus on Binary Classification**: Our focus on binary classification aligns with the real-world problem of calibrating radiologists' binary decisions about the presence of pathologies in medical images. Similarly, calibrating a language model’s confidence in its free-form answers involves binary classification of its correctness. This foundational case supports a wide range of practical applications, e.g., free-form language model generation.
>
> **Complexity of Our Approach**: To ensure clarity, it’s necessary that we provide concrete mathematical derivations of our method. To improve accessibility, we now supply code for estimating ECE, plotting reliability diagrams, and our proposed calibration method in supplementary material.
>
> **Scalability of Optimization**: While natural language allows for arbitrary certainty phrases, in practice, commonly used subsets are small. Furthermore, optimal transport solvers like the Sinkhorn algorithm scale efficiently to high-dimensional problems, ensuring practical usability.
>
> **Calibrating Humans**: Our study is the first to assess radiologists' calibration and provide tools to improve their calibration. We completely agree with the reviewer that improving human calibration is challenging and warrants further research. For instance, future work could explore how humans respond to different strategies for improving their calibration so that we can more effectively intervene. While we do not yet have definitive answers to the behavioral questions, we believe our work serves as an important first step toward addressing them.

---

> > ### Comment · Reviewer_qs5L · 2024-11-20
> >
> > Thanks for your reply! I concur with my initial assessment that this is a solid paper and am keeping my rating at 8.

---

> > > ### Author Response · Authors · 2024-11-24
> > > **response**
> > >
> > > Thanks for your helpful & encouraging review!

---

### Official Review · Reviewer_V58g · 2024-11-01

**Soundness:** 3
**Presentation:** 4
**Contribution:** 2
**Rating:** 6
**Confidence:** 4

**Summary:**

This paper introduces a novel miscalibration measure that incorporates a flexible, user-defined distribution for each bin, enabling a supervised "soft binning" approach. The authors propose this generalization to address the ambiguity in verbalized uncertainty levels, such as "unlikely" or "likely," which may not correspond to specific probability values. Additionally, the paper presents a post-hoc calibration method to adjust the use of uncertainty phrases based on the true distribution associated with each phrase. Experimental results on CT reports and outputs from language models demonstrate that the proposed miscalibration metric offers additional information of both human and language model usage of uncertainty phrases, and that the post-hoc calibration technique effectively refines the usage of these uncertainty phrases.

**Strengths:**

- Interesting proposal that uncertainty phrases could be associated with a distribution instead of a single confidence score.
- Very clear presentation of the method and insightful comparison against existing work.

**Weaknesses:**

- The link to uncertainty phrases seems superficial, as they serve merely as names for distributions without strong justification for this treatment.
- The motivation to represent uncertainty phrases as distributions is unclear; individual users often interpret these phrases in consistent orders, making scalar values (or simple binning) potentially sufficient. At a population level, it’s unclear why uncertainty phrases should align with complex empirical distributions rather than a straightforward rubric. In fact, it seems that using scalar values or distinct, non-overlapping probability regions might be a simpler and more operational approach.
- Limited Application: Only works for fixed uncertainty phrases in isolation, does not work with free-form generation with flexible uncertainty quantifiers.
- From table 1, the method does not seem to perform better than simpler post-hoc calibration methods that do not require confidence distribution.
- **NOTE** formula 6 is well outside the margin limit, please fix.

**Questions:**

- In the captions for Figure 4 and Table 1, the authors highlight the interpretability and informativeness of their calibration methods. However, the proposed post-hoc calibration method seems to make the calibration map less intuitive (e.g., Figure 3). Could the authors clarify how their methods enhance human interpretability, given the challenges of conceptualizing uncertainty as a distribution?
- It’s unclear how the ground truth uncertainty distribution is determined. The authors experimented with various methods to derive density functions for each confidence distribution, but it remains unclear which is best. Table 3 and Figure 5 suggest that the choice of ground truth density function impacts evaluation results. How can these ground truth density functions be reliably collected?
- In the "Calibrating Radiologists" experiments, confidence labels are extracted using Llama 3 (line 320 - 321), rather than from the radiologists' own report-level confidence labels. If this is correct, the analysis in Section 4.2 attributing uncertainty phrases to radiologists seems problematic. Could the authors clarify how this experiment was conducted?

---

> ### Author Response · Authors · 2024-11-19
> **response (part 1)**
>
> We appreciate the reviewer’s thoughtful feedback and provide clarifications below:
>
> **Motivation for Modeling Certainty Phrases as Distributions**: Certainty phrases are central to our study as they reflect how radiologists or language models communicate confidence. Representing certainty phrases as distributions is a modeling choice that captures their inherent variability and subjectivity. More concretely, modeling certainty phrases as distributions aligns with how individuals naturally express uncertainty. People rarely associate phrases like "Highly Likely" with specific probability values (e.g., 89%), but rather with ranges of confidence. In addition, doing so enables us to distinguish between phrases like "Exactly Equal Probability" and "Maybe", which might both mean roughly 50% probability but differ in their certainty (narrow vs. wide density function). Using distributions to represent certainty phrases is further supported by clear evidence from [survey](https://rad.bwh.harvard.edu/wp-content/uploads/2023/03/agreement-among-radiologists%EF%B9%96m1540397890itokaPmASEo6.jpg) of radiologists’ perception of different certainty phrases. Even if individuals are internally consistent, their interpretation of certainty phrases varies meaningfully across a population. For instance, "Possibly" was mapped to diverse probability ranges: 4.9% assigned it to (0-5%), 17.6% to (10-25%), 16.9% to (25-75%), and 0.7% to (75-90%), resulting in a population-level empirical distribution for “Possibly” that has non-uniform density and overlapping support with other empirical distributions (See Figure 1). This variation demonstrates why simple scalar values or binning are insufficient. Our framework naturally subsumes these simpler approaches as special cases (scalar values correspond to delta distributions, binning employs uniform distributions) while enabling us to model the rich structure we observe in how people (and thereby models) communicate uncertainty.
>
> **Clarify Limitations**: _Flexible certainty phrases_: Our method is not limited to a fixed set of phrases; we focus on commonly used ones for clarity. New phrases can be incorporated if their meanings can be quantified, for example, through human surveys or prompting language models. For instance, Appendix B.3 (_on-the-fly_ setup) demonstrates that our approach applies to LMs that can generate arbitrary Beta distributions to express their confidence rather than selecting from a fixed set of predefined phrases. _Free-form text_: Our approach handles free-form text by parsing (pathology, confidence) pairs from radiology reports, enabling calibration analysis for classifying specific pathologies. Calibration for arbitrary free-form text is not well-defined without framing it within some classification setup. For instance, in our LM experiments, we prompt models to generate free-form text and provide confidence ratings, enabling the analysis of calibration relative to model correctness in answering the question in context.
>
> **Comparison to Baseline Calibration Methods**: We apologize that our description of the experimental results in Table 1 did not make it sufficiently clear that our goal is not to significantly improve calibration of numerical scores (which is what Table 1 evaluates) but rather show that even if one reduces the output natural language expressions of certainty to scalar values (by simply taking the mean of the confidence distribution), our method remains competitive and does not compromise performance in terms of accuracy. We emphasize that our method's key advantage isn't in outperforming baseline calibration methods, but in directly operating on and producing natural language certainty phrases - a property that existing methods fundamentally lack. This makes our approach uniquely suited for improving human calibration in real-world settings, as it provides actionable guidance (e.g., suggesting radiologists use "May" instead of "Present" in their reporting to mitigate overconfidence) rather than abstract probability adjustments that can't be easily understood by humans. The goal is to maintain strong calibration performance while enabling calibration in natural language settings where traditional calibration methods cannot be meaningfully applied.
>
> **Latex Margin Overflow**: Thank you for catching it; we will correct it.

---

> ### Author Response · Authors · 2024-11-19
> **response (part 2)**
>
> **Informativeness of Reliability Diagram**: Figure 4 demonstrates that our approach produces smooth calibration curves that are easier to interpret. The deviation of the calibration curve (red) from the identity line, weighted by the score density (gray), provides an intuitive sense for the value of ECE. In contrast, binned reliability diagrams make it harder to visually assess calibration from calibration curves alone (see if you can guess which model is the strongest by just looking at calibration curves in Appendix C.3 figures). In addition, conclusions about model calibration from binned diagrams can vary significantly depending on the binning strategy used.
>
> **Interpretability of Calibration Method**: One goal of our post-hoc calibration method is to improve radiologists' calibration. Figure 3 shows how radiologists could achieve better calibration by following our method perfectly. We are conducting a follow-up clinical study to evaluate the method’s effectiveness in the real world. Our method is interpretable because it translates directly into actionable suggestions for radiologists, e.g., use “May” instead of “Present” and “Likely” in your reporting to mitigate overconfidence. In contrast, methods like histogram binning or Platt scaling are less practical, as they do not provide clear guidance on how radiologists can adjust their language in reporting to address miscalibration.
>
> **Confidence Distributions for LLM Experiments**: In Table 3 & Figure 5 (Figure 7 in updated draft), we found that predefining a fixed set of certainty phrases ($K=12$) for the model to choose from yields better calibration than alternatives. This fixed set of certainty phrases and their corresponding confidence distributions is generated by gpt-4o using 5th prompt in Table 5. These confidence distributions reflect the language model's interpretation of the associated phrases rather than being treated as definitive ground truth.
>
> **Certainty Phrases Preprocessing for Radiology Report Dataset**: We used Llama-3 to act as a “smart regex” to extract radiologists’ own report-level certainty phrases. As an example, when encountering “The opacity at lower right corner is likely pneumonia and possibly edema” in a radiology report, Llama-3 is prompted to extract (“Likely”, “Pneumonia”) and (“Possibly”, “Edema”).

---

> > ### Comment · Reviewer_V58g · 2024-11-26
> > **Response**
> >
> > Thank you for your response, my score remains as-is.

---

### Official Review · Reviewer_88Fm · 2024-11-03

**Soundness:** 3
**Presentation:** 2
**Contribution:** 2
**Rating:** 5
**Confidence:** 2

**Summary:**

This paper proposed a method to calibrate expressions of uncertainty in LLM generations. Instead of treating each uncertainty phrase as a probabilistic score between $[0, 1]$, this paper proposed that they be mapped to a distribution, expressed as a probability simplex. Under this formulation, calibration is reduced to a problem of learning an optimal transport. The authors conducted experiments over a curated dataset of radiologists, finding that their proposed method is comparable to baseline methods.

**Strengths:**

- By considering confidence as a probabilistic simplex, the formulation of calibration as an instance of optimal transport is interesting.

**Weaknesses:**

- <s>The experiments are performed over a self-curated dataset whose details are not aptly described. Maybe more details on the dataset is preferred. Additionally, there are public datasets that could be used for these experiments (e.g. https://nlp.jhu.edu/unli/, for natural language inference). As the authors proposed a general calibration method, I believe that datasets with diverse settings should be considered.</s>
 - Some experimental setup are unclear. See below.
 - Table 1: The proposed method's improvement is marginal, or even non-existent as compared to baseline calibration methods such as Platt scaling and histogram binning. This does not justify the usefulness of your proposed method.

**Questions:**

- L265: $\triangle_{K-1}$: The general notation for the probabilistic simplex should be $\Delta^{K-1}$.
 - L420: I am dubious of LLMs' ability to generate a sequence like `Beta(2, 3)` that accurately reflects its internal belief. How many exemplars are used in the in-context learning? This is not spelled out clearly in the Appendix. (It seems that only 1 example is used?)

---

> ### Author Response · Authors · 2024-11-19
> **response**
>
> We appreciate the reviewer’s thoughtful feedback and provide clarifications below:
>
> **Curated Dataset Details**:
> We have added additional details about our curated dataset in Appendix B.1.
>
> **Evaluation on Public Datasets such as UNLI**:
>
> We appreciate the reviewer bringing UNLI to our attention. While UNLI includes human-provided probability annotations, these are not necessary for evaluating a language model's calibration when it outputs certainty phrases about an answer's correctness—ground-truth labels suffice for such evaluations. Moreover, our existing evaluations on public datasets SciQ (where language models are well-calibrated) and TruthfulQA (where calibration is poor) already provide complementary insights into model calibration across diverse settings. As such, UNLI does not add to the diversity of setups we already investigate in the paper.
>
> If the reviewer’s suggestion was to evaluate human annotators’ calibration, we note that the UNLI annotations are scalar probabilities rather than certainty phrases, which means they do not provide sufficient information to evaluate the utility of our method designed for calibrating certainty phrases. Consequently, we believe UNLI does not provide additional information beyond what is already captured by the datasets we have used. That said, we welcome the reviewer’s suggestions for additional datasets that could provide new insights into calibrating natural language expressions of (un)certainty.
>
> **Comparison to Baseline Calibration Methods**: We apologize that our description of the experimental results in Table 1 did not make it sufficiently clear that our goal is not to significantly improve calibration of numerical scores (which is what Table 1 evaluates) but rather show that even if one reduces the output natural language expressions of certainty to scalar values (by simply taking the mean of the confidence distribution), our method remains competitive and does not compromise performance in terms of accuracy. We emphasize that our method's key advantage isn't in outperforming baseline calibration methods, but in directly operating on and producing natural language certainty phrases - a property that existing methods fundamentally lack. This makes our approach uniquely suited for improving human calibration in real-world settings, as it provides actionable guidance (e.g., suggesting radiologists use "May" instead of "Present" in their reporting to mitigate overconfidence) rather than abstract probability adjustments that can't be easily understood by humans. The goal is to maintain strong calibration performance while enabling calibration in natural language settings where traditional calibration methods cannot be meaningfully applied.
>
> **Probability Simplex Notation**: Thank you for catching this typo; we will correct it.
>
> **Instruction Template & Reproducibility of LM Results**: We clarify that we don’t use in-context learning but rather a zero-shot instruction template to elicit verbalized confidence by predicting beta distribution parameters directly (in Table 5). The code for replicating these experiments is now provided as supplementary material, and we encourage the reviewer to verify its reproducibility.

---

> > ### Comment · Reviewer_88Fm · 2024-11-28
> >
> > Thanks for the explanation on the experimental setup. I am willing to increase the score to 5.

---

### Author Response · Authors · 2024-11-19
**shared response**

We sincerely thank all reviewers for their time and helpful comments, which have helped us strengthen our submission. We are happy they found our paper (1) offers clear and rigorous presentation of method and insightful comparison against existing work, (2) provides an interesting proposal to treat certainty phrases as distributions and formulate calibration as optimal transport, and (3) demonstrates strong real-world applicability (e.g., in medicine).

In response to reviewers’ comments, we have updated the draft as summarized in the table below.

| Changes                                                                                                                                                                                 | Reviewer         |
| --------------------------------------------------------------------------------------------------------------------------------------------------------------------------------------- | ---------------- |
| Provide code to reproduce LM experiments in the supplementary material.                                                                                                                 | 88Fm,qs5L       |
| Clarify interpretation of results that compares our approach with baselines (Table 1).                                                                                                  | 88Fm,V58g,itYd   |
| Add future work to conduct clinical study on whether radiologists can adapt their use of certainty phrases to improve calibration (Section 5).                                          | V58g,qs5L,itYd |
| Add more details on the self-curated radiology report dataset (Appendix B.1).                                                                                                           | 88Fm             |
| Add qualitative examples on how radiologists and LMs use certainty phrases (Appendix B.2).                                                                                              | itYd             |
| Clarify why we believe our smooth calibration curves are more informative than previously used methods (Appendix C.3) and the interpretability of our calibration method (Section 4.2). | V58g             |
| Add detailed answers on why we treat certainty phrases as distributions, and our methods' use cases (Appendix D).                                                                       | V58g             |
| Fix typos, formatting issues, & unclear sentences in experimental setups.                                                                                                               | 88Fm,V58g       |

Beyond these changes, we have addressed all comments raised by the reviewers (see individual responses below). Again we deeply appreciate the feedback and are happy to address any remaining questions during the discussion phase.

---

### Meta-Review · Area_Chair_JntK · 2024-12-21

**Metareview:**

This paper explores expressions of certainty ("maybe" and "likely") and how to calibrate LLM certainty with these expressions.  Rather than pegging these to a particular likelihood value, this paper seeks to understand both individual- and population-level variation in the use of these phrases. Instead of representing confidence as a score, this paper represents it as a distribution over [0,1]. The paper generalizes notions of ECE to this setting and connects this to kernel density estimation. With everything in a continuous space, calibration can be viewed as optimal transport.  The paper then evaluates how calibrated radiologists are on their predictions, as well as calibration of LLMs with probabilistic and verbal calibration. Post-hoc calibration based on optimal transport performs similarly to baselines, although this is based on treating each certainty phrase as the mean of its distribution.

This paper presents an interesting new way of thinking about expressions of certainty. It is well-written and develops a new mathematical framework for thinking about this. In my view, this could significantly impact the ongoing work of researchers in this area, although more work and more data is needed to really deliver on this view of calibration.

The main issues here are the lack of a slam-dunk experimental result here. The gains reported in Table 1 are marginal, and the authors backpedal and say that improving things here is not their intent. I see this claim and agree that it is hard to show why this framework is worth using in an empirical way, but also feel that this paper should show something stronger experimentally.  There are some other weaknesses such as the self-curated nature of the dataset and the overall framing of the work.

While the weaknesses here are significant, I think the significant recent body of work on linguistic calibration would benefit from this treatment. This paper breaks new ground in how to think about and model this task which other researchers can substantially benefit from. There are rough edges in the paper, but it is thought provoking and likely to be more impactful than papers that marginally advance SOTA in some area.

**Additional Comments On Reviewer Discussion:**

88Fm brings up the self-curated nature of the dataset. Although the authors address this and the discussion of UNLI, I think the bigger issue here is lack of connection to established datasets in the literature, which remains som
ewhat unresolved as a point.

88Fm, V58g, and itYd also critiques the gains as marginal. The authors push back and state that their intent is to "directly operat[e] on and produc[e] natural language certainty phrases - a property that existing methods fundamentally lack". While this is true, it remains the case that the paper doesn't show something very strong experimentally.

V58g asks about the fundamental nature of certain phrases as being distributions. I am convinced by what the authors say conceptually here.

---

> ### Public Comment · ~Peiqi_Wang2 · 2025-02-06
> **response**
>
> We sincerely appreciate the time and effort the reviewer and meta-reviewer have put into evaluating our work. We acknowledge that a slam-dumk experiment demonstrating the full potential of our approach is a lingering issue. As part of our ongoing work, we are actively applying this method to clinical problems/datasets, where we believe it can have significant impact. We hope this paper serves as an important first step in that direction and contributes to the broader discussion on LLM calibration. Thank you again for your thoughtful feedback!

---

### Decision · Program_Chairs · 2025-01-22

Accept (Poster)